# nvBench 2.0: Resolving Ambiguity in Text-to-Visualization through Stepwise Reasoning

**Tianqi Luo**[1] **Chuhan Huang**[1] **Leixian Shen**[2] **Boyan Li**[1] **Shuyu Shen**[1]
**Wei Zeng**[1,2] **Nan Tang**[1,2] **Yuyu Luo**[1,2*]

[1]The Hong Kong University of Science and Technology (Guangzhou), Guangzhou, China
[2]The Hong Kong University of Science and Technology, Hong Kong SAR, China

## Abstract

Text-to-Visualization (*Text2VIS*) enables users to create visualizations from natural language queries, making data insights more accessible. However, *Text2VIS* faces challenges in interpreting ambiguous queries, as users often express their visualization needs in imprecise language. To address this challenge, we introduce nvBench 2.0, a new benchmark designed to evaluate *Text2VIS* systems in scenarios involving ambiguous queries. nvBench 2.0 includes 7,878 natural language queries and 24,076 corresponding visualizations, derived from 780 tables across 153 domains. It is built using a controlled ambiguity-injection pipeline that generates ambiguous queries through a reverse-generation workflow. By starting with unambiguous seed visualizations and selectively injecting ambiguities, the pipeline yields multiple valid interpretations for each query, with each ambiguous query traceable to its corresponding visualization through step-wise reasoning paths. We evaluate various Large Language Models (LLMs) on their ability to perform ambiguous *Text2VIS* tasks using nvBench 2.0. We also propose Step-Text2Vis, an LLM-based model trained on nvBench 2.0, which enhances performance in ambiguous scenarios through step-wise preference optimization. Our results show that Step-Text2Vis outperforms all baselines, setting a new state-of-the-art for ambiguous *Text2VIS* tasks. Our source code and data are available at `https://nvbench2.github.io/`.

## 1 Introduction

Text-to-Visualization (*Text2VIS*) democratizes data exploration and analysis by enabling users to generate Visualizations (vis) from text queries [1–5]. While recent advances in Large Language Models (LLMs) [6–14] have significantly enhanced translation accuracy, they struggle with a fundamental challenge: *text query ambiguity*—a single query often maps to multiple valid visualizations, each representing a different interpretation of the user's intent [15–22].

In *Text2VIS*, there are two layers of ambiguity: the **data layer**, which governs how a query selects, filters, and transforms data, and the **visualization layer**, which determines how the data is visually represented. For example, in Figure 1, the text query "Show the gross trend of comedy and action movies by year" appears straightforward but contains multiple ambiguities. At the data layer, "gross" could mean either `World_Gross` or `Local_Gross` column, while "comedy and action" implicitly requires filtering the column `Genre`. Moreover, "by year" implies temporal binning on `Date` with aggregation on "gross", neither of which is explicitly specified. At the visualization layer, "trend" might suggest a bar or line chart, with the x-channel representing the binned temporal data, the y-channel showing the aggregated "gross" values, and the implicit

---

*Yuyu Luo is the corresponding author (yuyuluo@hkust-gz.edu.cn)

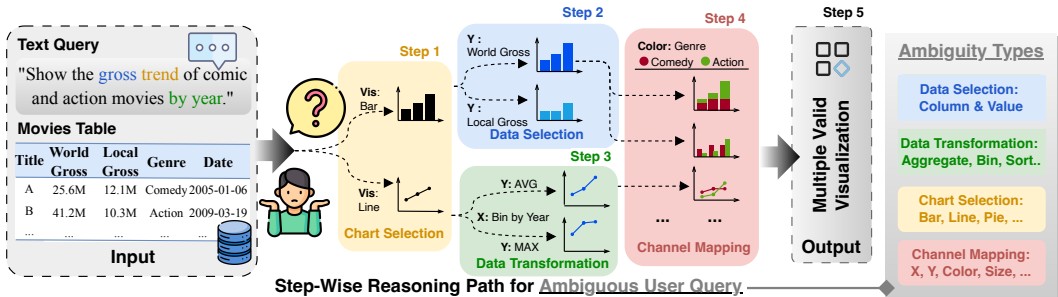

Figure 1: Example of reasoning appropriate visualizations from an ambiguous query.

column `Genre` mapped to the `color-channel`, where comedy and action are represented by distinct colors in the visualization. This example highlights how ambiguities at both the data and visualization layers interact, complicating the mapping from text queries to visualizations.

**Existing Benchmarks and Their Limitations.** Although several benchmarks for the *Text2VIS* task exist [23, 24, 6, 25–28], as shown in Table 1, none explicitly evaluates how the *Text2VIS* systems handle ambiguity. In fact, existing efforts [24, 27, 6] often overlook this issue by adhering to the *single-correct-answer* paradigm, where each text query maps to exactly one valid visualization. For example, nvBench [24] maps a text query to a unique visualization, ignoring more than 60% of real-world ambiguous cases [25]. Similarly, Dial-NVBench [27] supports multi-turn clarification but assumes that the final query is well-specified, which sidesteps the inherent ambiguities in user intents.

This narrow focus leaves a critical gap in the push to advance *Text2VIS* systems. *How can we evaluate and improve their ability to generate valid visualizations from ambiguous text queries?*

**Design Considerations.** To address this challenge, a benchmark is needed that tests *Text2VIS* solutions on handling ambiguous text queries, recognizing multiple valid interpretations, and providing an appropriate set of answers. This benchmark should include diverse ambiguous queries, multiple valid outputs, reasoning paths explaining the ambiguity, and broad domain coverage.

**Our Proposal.** To fill this gap, we propose nvBench 2.0, the first benchmark for generating visualizations from ambiguous text queries (*i.e.,* the ambiguous *Text2VIS* task). This dataset provides a robust foundation for evaluating *Text2VIS* solutions in scenarios where text query ambiguity is a key challenge. nvBench 2.0 includes 7,878 text queries and 24,076 corresponding visualizations, derived from 780 tables across 153 domains. It meets the design considerations through a controllable ambiguity-injected *Text2VIS* data synthesis pipeline. This pipeline uses a reverse-generation workflow, starting with a seed visualization and injecting ambiguity to create multiple valid interpretations. We then generate an ambiguous text query for each set of modified visualizations, incorporating injected ambiguity. The pipeline's transparency allows tracking of how interpretations lead to distinct visualizations, supported by reasoning paths detailing the ambiguity resolution process. This traceability enables researchers to assess the effectiveness and interpretability of ambiguity resolution, ensuring an accurate and explainable process.

**Contributions.** Our main contributions are summarized as follows:

- **Ambiguity-Injected Data Synthesizer.** We develop a *Text2VIS* data synthesizer that generates ambiguous data by selectively injecting ambiguities into seed visualizations, yielding multiple valid answers for each text query while providing step-wise disambiguation reasoning paths. (Section 2.1)
- **nvBench 2.0 Benchmark.** We present nvBench 2.0, the first benchmark designed for the ambiguous *Text2VIS* task. It contains 7,878 text queries and 24,076 corresponding visualizations, derived from 780 tables across 153 domains. Each *Text2VIS* sample is paired with a disambiguation reasoning path, providing clear explanations of how the ambiguity is resolved and ensuring the interpretability of the ambiguity resolution process. (Section 2.2)
- **Step-Text2Vis for Ambiguous *Text2VIS* Tasks.** We propose Step-Text2Vis, an LLM-based model trained on nvBench 2.0. By leveraging step-wise preference optimization and the provided reasoning paths, Step-Text2Vis model achieves the highest F1@3 (81.50%) and F1@5 (80.88%), outperforming prompting GPT-4o by 22.54% and 21.85%, respectively. (Section 3)
- **Extensive Evaluation.** We conduct comprehensive experiments to validate the effectiveness of nvBench 2.0 for training and evaluating *Text2VIS* systems under ambiguity. Our findings reveal the

Table 1: Comparison of *Text2VIS* benchmarks.

| Datasets | #-Tables | #-Samples | | TEXT → VIS Mapping | TEXT Ambiguity | Reasoning Paths | TEXT Generation |
| | | #-VIS | #-TEXT | | | | |
|---|---|---|---|---|---|---|---|
| Quda [26] | 36 | - | 14035 | - | ✦ | ✗ | Human-based |
| NLV [25] | 3 | 30 | 814 | $n \rightarrow 1$ | ✦ | ✗ | Human-based |
| Dial-nvBench [27] | 780 | 7247 | 124449 | $n \rightarrow 1$ | ✗ | ✗ | Rule-based |
| VL2NL [23] | 1981 | 1981 | 3962 | $1 \rightarrow 1$ | ✦ | ✗ | LLM-based |
| VisEval [6] | 748 | 2524 | 1150 | $1 \rightarrow 1$ | ✗ | ✗ | LLM-based |
| nvBench [24] | 780 | 7247 | 25750 | $1 \rightarrow 1$ | ✗ | ✗ | Rule-based |
| **nvBench 2.0** | 780 | 24076 | 7878 | $1 \rightarrow n$ | ✓ | ✓ | LLM-based |

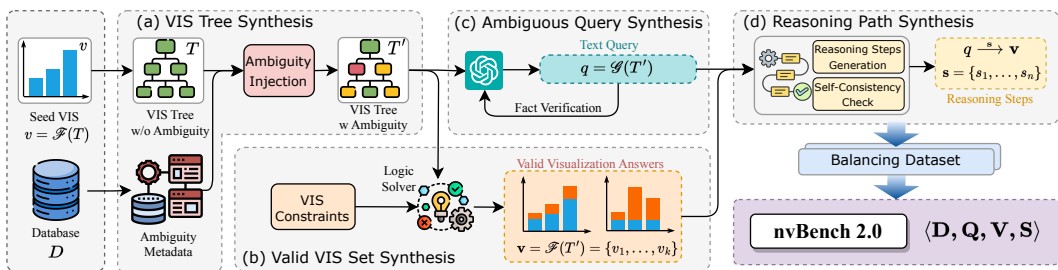

Figure 2: The Pipeline for Synthesizing nvBench 2.0.

limitations of existing models when faced with ambiguous queries while demonstrating that the Step-Text2Vis model outperforms baseline approaches and achieves state-of-the-art performance in ambiguous *Text2VIS* tasks. (Section 4)

## 2 nvBench 2.0

In this section, we will first elaborate on how to develop nvBench 2.0 with an Ambiguity-Injected Data Synthesizer (Section 2.1), and then describe the characteristics of nvBench 2.0 (Section 2.2).

### 2.1 Ambiguity-Injected *Text2VIS* Data Synthesizer

Figure 2 provides a high-level overview of our *Ambiguity-Injected* Text2VIS *Data Synthesizer*. The pipeline begins with a data table $D$ and an unambiguous seed VIS $v$, from which a VIS tree $T$ (without ambiguity) is derived. Through a systematic ambiguity injection process, this tree is transformed into $T'$ (with ambiguity), serving as intermediate products in the workflow. Subsequently, the pipeline generates an ambiguous text query $q$ alongside a corresponding set of valid VIS $\mathbf{v} = \{v_1, \ldots, v_k\}$, while also producing step-wise reasoning paths $\mathbf{s} = \{s_1, \ldots, s_k\}$ for each valid VIS. The final output of the pipeline is nvBench 2.0, structured in the form $\langle D, Q, V, S \rangle$. Next, we will go through our pipeline step by step.

**Data Preparation.** To construct nvBench 2.0, we obtained data tables from nvBench 1.0 [24] and BIRD [29], ensuring broad coverage of the domain and relevance in the real world. We retained select seed visualizations from [24] while introducing new ones to address the limited variety of chart types. Novel chart types, such as boxplots and heatmaps, and additional encoding channels, like "size" (enabling scatterplots to transition to bubble charts), were incorporated to enhance visual encoding diversity and expressiveness.

**Step 1: Ambiguity-aware VIS Tree Synthesis.** We start the process by building an initial VIS tree $T$ from a seed VIS $v$. The tree $T$, structured as an Abstract Syntax Tree (AST) based on VIS grammar, encodes components such as data mappings, mark types, and encoding channels. Each node in $T$ represents a component in VIS and whether it has ambiguity. We then extract ambiguity metadata from the database $D$ using a structured knowledge graph (KG). This involves a KG-driven semantic alias identification process and an LLM-driven refinement process. Metadata systematically categorizes semantic ambiguities within the table schema to guide the creation of ambiguity-aware VIS trees.

Subsequently, we transform $T$ into an ambiguity-aware tree $T'$ through a controlled ambiguity injection process. This involves two operations: (1) *injecting ambiguous nodes* to add semantically ambiguous components, (2) *injecting implicit nodes* to replace fixed components with a blank placeholder, or to add unspecified but essential components for the altered VIS tree. This injection process ensures that $T'$ can branch into multiple valid interpretations, capturing the full spectrum of possible outcomes for an ambiguous text query.

**Step 2: Valid VIS Set Synthesis.** The partially ambiguous VIS tree $T'$ is processed through an Answer Set Programming (ASP) solver [30], which applies grammar constraints to transform the ambiguous tree into a resolved VIS set $\mathbf{v} = \{v_1, \ldots, v_k\}$. The number of resulting VIS, $k = |\mathbf{v}|$, indicates the ambiguity level—how many distinct interpretations the solver deems valid for given $T$.

The completeness of a valid answer set is ensured by the ASP solver's exhaustive enumeration of all stable models satisfying the encoded VIS constraints through its declarative logic programming framework [30]. This guarantees that all possible answers consistent with the input text query and grammar constraints are generated, providing a comprehensive set of solutions for ambiguous queries. Please refer to Appendix B.2 for more details.

**Step 3: Ambiguous Query Synthesis.** We leverage an LLM-based `Query Generator` to synthesize an ambiguous text query $q$ for each modified VIS tree $T'$, incorporating the newly introduced ambiguities into a single query. This approach ensures that every synthesized valid answer in set $\mathbf{v} = \{v_1, \ldots, v_k\}$ faithfully represents the ambiguous intents of the query. Finally, an LLM-based `Query Verifier` checks consistency, confirming that the final text query accurately reflects all valid answers. Please refer to Section B.3 in the Appendix for more details.

**Step 4: Ambiguity-resolved Reasoning Paths Synthesis.** Finally, we generate *stepwise disambiguation reasoning paths* to guide the resolution of each ambiguity in producing every valid VIS. These paths are built using an LLM with an automated self-consistency validation mechanism to ensure accuracy. By systematically extracting and articulating the discrete reasoning steps from the initial text query $q$ to the set of VIS $\mathbf{v}$, we provide a transparent and comprehensive explanation of how each query maps to its corresponding VIS outcomes. Details are provided in Section B.4.

**Dataset Balancing and Quality Control.** We implement several strategies to ensure the high quality of our nvBench 2.0. (1) *Ambiguity Level Regulation*: We constrain the ambiguity level to $k \leq 5$, ensuring that each retained sample illustrates a meaningfully different way of interpreting the partially ambiguous tree. (2) *Visualization Diversity*: From a large pool of randomly generated seed visualizations and ambiguity-injected trees, we compute pairwise distances and select the most diverse subset to enhance the variety of visualizations. (3) *Query Verification*: In the query synthesis process, although GPT-4o effectively captures most ambiguous nodes in $T'$, it introduces unintended facts in approximately 5% of generated queries. We implement a fact verification process to identify and refine these unwanted intents. We then conduct manual reviews by two postgraduate students to re-check the identified queries. This pipeline ensures that each ambiguous aspect in VIS tree synthesis phase is clearly reflected in the final mapping of *Text2VIS*, allowing researchers and practitioners to evaluate how effectively models handle and explain different interpretations.

## 2.2 nvBench 2.0 Characteristics

This section describes the key characteristics of nvBench 2.0, focusing on the interaction of ambiguity types, levels, patterns, query styles, and visualizations.

**Ambiguity Types and Levels.** An important contribution of nvBench 2.0 is the systematic introduction of controlled ambiguity types and ambiguity levels. Figure 3 (a) categorizes ambiguity by type: Data Transformation (DT) ambiguities are most prevalent (50.55%), followed by Channel Mapping (CM) ambiguities (23.30%), with Data Selection (DS) and Chart Type Selection (CT) ambiguities represented by 16.10% and 10.00%, respectively. As shown in Figure 3 (b), the majority of samples (44.10%) have an ambiguity level of 2, indicating that two valid visualizations exist for each text query. nvBench 2.0 also contains a substantial number of samples with ambiguity levels of 3, 4, and 5, enabling a thorough evaluation of systems under increasingly complex ambiguous scenarios.

**Ambiguity Combination Patterns.** Since multiple ambiguity types can occur in a single data sample, Table 2 shows the most frequent ambiguity combination patterns. The most common pattern is CM+DT (2,190 instances), followed by CM+DS (486), while more complex multi-category

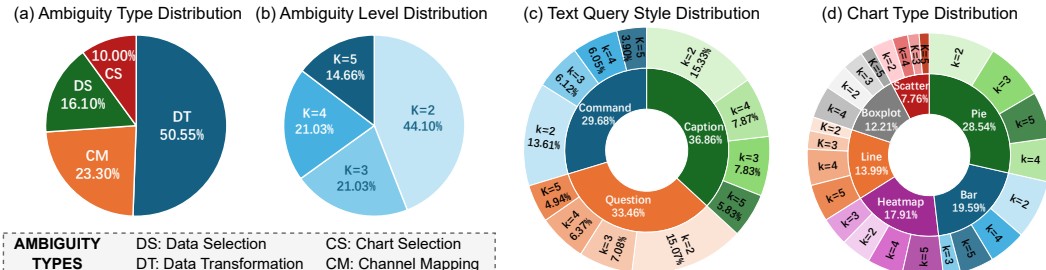

Figure 3: Key Statistics of nvBench 2.0.

Table 2: Statistics of Ambiguity Combination Patterns

| Pattern | Count | Pattern | Count |
|---------|-------|---------|-------|
| CM+DT | 2190 | CM+DS | 486 |
| CM+DS+DT | 364 | CM+CS+DT | 171 |
| CM+CS | 158 | CS+DT | 34 |

ambiguities are less frequent. Channel Mapping and Data transformation ambiguities often co-occur with other ambiguity types. This distribution underscores the challenge of resolving overlapping ambiguity types in vis generation tasks.

**Text Queries.** Figure 3(c) presents the query style distribution. We balance the distribution to follow the observation in work [25], where command-based queries are most frequent, question-based and caption-like queries are also useful but less frequent. Appendix C provides a detailed breakdown of query styles across different chart types, along with word count statistics.

**Visualizations.** Figure 3(d) illustrates the distribution of the chart types. nvBench 2.0 comprises six chart types, with Bar and Pie being the most prevalent, aligning their common use in the real world. Other chart types cater to more specialized analytical purposes, aligning with typical vis practices. Notably, the ambiguity levels are distributed similarly across all chart types, ensuring a well-balanced data distribution to evaluate *Text2VIS* systems under varying degrees of ambiguity.

## 3 Step-Text2Vis for Ambiguous *Text2VIS*

In this section, we present Step-Text2Vis, a new model for the ambiguous *Text2VIS* task. Step-Text2Vis addresses ambiguity by incorporating a step-wise reasoning process, as detailed in Section B.4, and leveraging the rich step-wise data provided by nvBench 2.0. Built on base LLMs, Step-Text2Vis is fine-tuned on nvBench 2.0 using a pipeline that aligns its outputs with the dataset's reasoning paths via supervised fine-tuning and step-wise preference optimization (Step-DPO) [31].

### 3.1 Preference Optimization with Step-DPO

Previous *Text2VIS* methods have typically employed either prompting LLMs [32] or fine-tuning LLMs [27], where the LLM is directly tasked with generating the final vis definition based on text and table schema information.

Recently, process supervision paradigms [33] and preference optimization techniques [31] have demonstrated significant advancements across various domain tasks. A pivotal aspect in validating the effectiveness of nvBench 2.0 is determining how to leverage the step-wise disambiguation reasoning paths within the nvBench 2.0 dataset to provide process supervision and enhance model performance. Consequently, we adopt the Step-DPO [31], which utilizes step-wise paired correct and incorrect samples for preference optimization, thereby delivering rich process supervision signals to the model and fostering improved accuracy at each step.

Formally, we define an input prompt $x$ and an vis answer $y$, where $x$ includes text and table schema, and $y$ can be represented as $s_1 \oplus \cdots \oplus s_n$, where $s_i$ denotes the $i$-th reasoning step defined in Section B.4. Given the input $x$ and a sequence of correct preceding reasoning steps

$s_{1\sim k-1} = s_1 \oplus \cdots \oplus s_{k-1}$, Step-DPO aims to maximize the probability of the correct next reasoning step $s_{win}$ and minimize the probability of the incorrect one $s_{lose}$. This objective can be formulated as:

$$\mathcal{L}(\theta) = -\mathbb{E}_{(x, s_{1\sim k-1}, s_{win}, s_{lose}) \sim D_p}$$
$$\left[ \log \sigma \left( \beta \log \frac{\pi_\theta(s_{win}|x, s_{1\sim k-1})}{\pi_{ref}(s_{win}|x, s_{1\sim k-1})} - \beta \log \frac{\pi_\theta(s_{lose}|x, s_{1\sim k-1})}{\pi_{ref}(s_{lose}|x, s_{1\sim k-1})} \right) \right] \quad (1)$$

where $D_p$ represents a step-wise preference dataset. $\pi_\theta(\cdot|x, s_{1\sim k-1})$ denotes the policy model to be optimized, while $\pi_{ref}(\cdot|x, s_{1\sim k-1})$ refers to the reference model, which remains unchanged during the training process. The hyperparameter $\beta$ controls the divergence between the optimized policy and the reference model.

## 3.2 Cold-start with Supervised Fine-tuning

Prior studies, such as those employing Chain-of-Thought (CoT) [34] prompting, have demonstrated the capability of LLMs to engage in step-wise reasoning through the utilization of simple "`think step-by-step`" instructions. However, under this paradigm, the planning of steps and the format of output are indiscriminate. This poses challenges in the precise extraction of answers corresponding to each individual step, and consequently, impedes the accurate alignment with the step-wise data provided within the nvBench 2.0 dataset for the purpose of validating step-level correctness. To address this limitation, we use nvBench 2.0 training set and employ Supervised Fine-Tuning (SFT) as a cold-start mechanism to facilitate the LLM's learning of our predefined step-wise output format. The specific training setup and prompt templates are in Appendix D.

## 3.3 Step-wise Preference Data Construction

A crucial aspect of Step-DPO is the acquisition of a step-wise preference dataset. As described in Section 3.1, our nvBench 2.0 dataset contains step-wise ground-truth. Therefore, we adopt an online data collection strategy. Initially, we utilize a model that has undergone Supervised Fine-Tuning (SFT) cold-start to perform inference on the nvBench 2.0 development set, yielding $D_0 = \{(x, \hat{y})\}$, where $\hat{y}$ represents the model's step-wise output, expressible as $\hat{s}_1 \oplus \cdots \oplus \hat{s}_n$. Subsequently, we conduct a step-wise evaluation comparing $\hat{y}$ with the ground-truth $y$, verifying the correctness of each step until the identification of the first error, and recording its corresponding step number $k$. We designate the erroneous step $\hat{s}_k$ as the incorrect reasoning step $s_{lose}$, and the ground-truth step $s_k$ as the correct reasoning step $s_{win}$. The construction of the preference dataset $D_p = \{(x, \hat{s}_{1\sim k-1}, s_{win}, s_{lose}\}$ is then readily achieved through the integration of input $x$ and previous reasoning steps $\hat{s}_{1\sim k-1}$.

# 4 Experiments

In our experiments, we aim to answer two fundamental questions about ambiguous *Text2VIS* tasks. First, how effectively do different approaches—including state-of-the-art LLMs and our proposed Step-Text2Vis —handle vis generation from text queries with varying levels of ambiguity? Second, what impact does step-wise reasoning have on performance across different chart types and ambiguity scenarios compared to direct generation approaches? To address these questions, we designed a comprehensive evaluation framework comparing different methods with or without stepwise reasoning. We assess performance using standard information retrieval metrics across multiple levels of ambiguity.

## 4.1 Experimental Setup

**Datasets.** We use nvBench 2.0 for our experiments and randomly divide the data set into training, development, and testing sets in a ratio of 80%, 10%, and 10%, containing 6377, 750, and 751 samples, respectively.

**Methods.** We evaluate the performance on ambiguous *Text2VIS* tasks using both prompting-based and fine-tuning-based methods with nvBench 2.0. The primary goal is to assess the model's ability to generate diverse and semantically accurate visualizations in response to ambiguous text queries. Please refer to Section D in the Appendix for more details.

Table 3: Overall performance comparison between different models on nvBench 2.0. The table presents Recall@K, Precision@K, and F1@K metrics across different model families. Rows 1–12 shows results for models using prompting-based method. The last two rows, Qwen2.5-7B-SFT and Step-Text2Vis (ours), are models using supervised fine-tuning method and preference learning method with optimization on nvBench 2.0. **Bold values** indicate the best performance for each metric.

| Model | Recall@K(%) | | | Precision@K(%) | | | F1@K(%) | | |
|---|---|---|---|---|---|---|---|---|---|
| | K=1 | K=3 | K=5 | K=1 | K=3 | K=5 | K=1 | K=3 | K=5 |
| GPT-4o-mini | 34.72 | 51.92 | 54.65 | 91.88 | 86.86 | 81.76 | 49.31 | 59.73 | 57.60 |
| GPT-4o | 36.56 | 46.35 | 46.79 | 97.07 | 95.83 | **95.52** | 51.96 | 58.96 | 59.03 |
| Claude-3.5-Haiku | 36.03 | 67.95 | 67.95 | 95.74 | 93.92 | 93.83 | 51.22 | 75.63 | 75.56 |
| Qwen2.5-7B | 34.65 | 46.20 | 47.17 | 92.68 | 90.68 | 89.33 | 49.34 | 57.09 | 56.67 |
| Qwen3-235B | 29.37 | 55.59 | 59.39 | 78.83 | 72.70 | 64.84 | 41.87 | 58.77 | 55.39 |
| GPT-4o-mini-Step | 35.13 | 47.68 | 47.91 | 93.48 | 92.54 | 92.08 | 49.96 | 59.29 | 59.10 |
| GPT-4o-Step | 36.30 | 48.92 | 49.21 | 96.94 | 95.47 | 95.08 | 51.72 | 60.78 | 60.66 |
| Claude-3.5-Haiku-Step | 35.70 | 65.38 | 65.52 | 94.67 | 92.23 | 91.97 | 50.75 | 72.84 | 72.75 |
| Qwen2.5-7B-Step | 35.20 | 61.86 | 64.08 | 93.61 | 89.26 | 86.23 | 50.05 | 68.56 | 67.76 |
| Qwen3-235B-Step | **37.49** | 72.83 | 75.39 | **99.60** | **95.52** | 92.21 | **53.29** | 78.62 | 77.78 |
| Qwen2.5-7B-SFT | 33.23 | 73.44 | 76.32 | 88.42 | 83.36 | 80.18 | 47.26 | 75.79 | 75.30 |
| **Step-Text2Vis (ours)** | 37.30 | **77.09** | **79.74** | 99.20 | 94.27 | 91.17 | 53.04 | **81.50** | **80.88** |

*Prompting-based Methods.* We evaluate two prompting strategies: Direct Prompting and Step Prompting. In Direct Prompting, the model receives structured *Data Schema Information* and a *text query* as input, subsequently generating 1-5 distinct visualizations to cover possible interpretations of the ambiguous query. This strategy is applied to the **GPT-4o-mini**, **GPT-4o**, **Claude-3.5-Haiku**, **Qwen2.5-7B** and **Qwen3-235B** models. For Step Prompting, models are guided to "`think step-by-step`", explicitly articulating their reasoning process before generating VIS. Models utilizing this approach are denoted by a suffix "-Step" (e.g., **GPT-4o-mini-Step**, **GPT-4o-Step**, **Claude-3.5-Haiku-Step**, **Qwen2.5-7B-Step**, **Qwen3-235B-Step**)

*Supervised Fine-tuning Method.* We performed supervised fine-tuning on the Qwen2.5-7B-Instruct model, resulting in a baseline model named **Qwen2.5-7B-SFT**. This model was trained using the standard SFT approach in the training set, enabling the direct generation of multiple VIS answers for the interpretation of ambiguities.

*Preference Learning Method.* We developed an advanced model, referred to as **Step-Text2Vis**, designed to handle ambiguity in *Text2VIS* through step-wise reasoning as detailed in Section 3. Following an initial supervised fine-tuning, we constructed a preference dataset from the nvBench 2.0 development set specifically for the preference training of Step-Text2Vis.

**Evaluation Metrics.** Following prior work [35, 36, 24], we adopt the following metrics: **Precision@K (P@K)**: Measures the proportion of valid VIS in the top K output, reflecting the precision of the recommendation. **Recall@K (R@K)**: Evaluates the proportion of valid VIS identified, indicating coverage of the golden VIS space. **F1@K**: Balances precision and recall, ensuring both high accuracy and comprehensive coverage. All metrics are reported for $K \in 1, 3, 5$ to assess performance across varying recommendation set sizes.

## 4.2 Experimental Results

**Overall Results.** Table 3 presents the comprehensive performance evaluation of different models on nvBench 2.0. Our proposed Step-Text2Vis achieves state-of-the-art performance across most metrics, significantly outperforming both prompting-based and fine-tuning-based baselines. Specifically, Step-Text2Vis obtains the highest F1@3 (81.50%) and F1@5 (80.88%), demonstrating its superior ability to handle ambiguity in *Text2VIS* tasks. Step-wise reasoning consistently improves performance across most models, although the benefits vary by model architecture. While some models achieve higher recall or precision in specific scenarios, they fail to maintain competitive F1 scores, indicating an imbalance between precision and recall. For GPT-4o, Qwen2.5-7B, and Qwen-3-235B models,

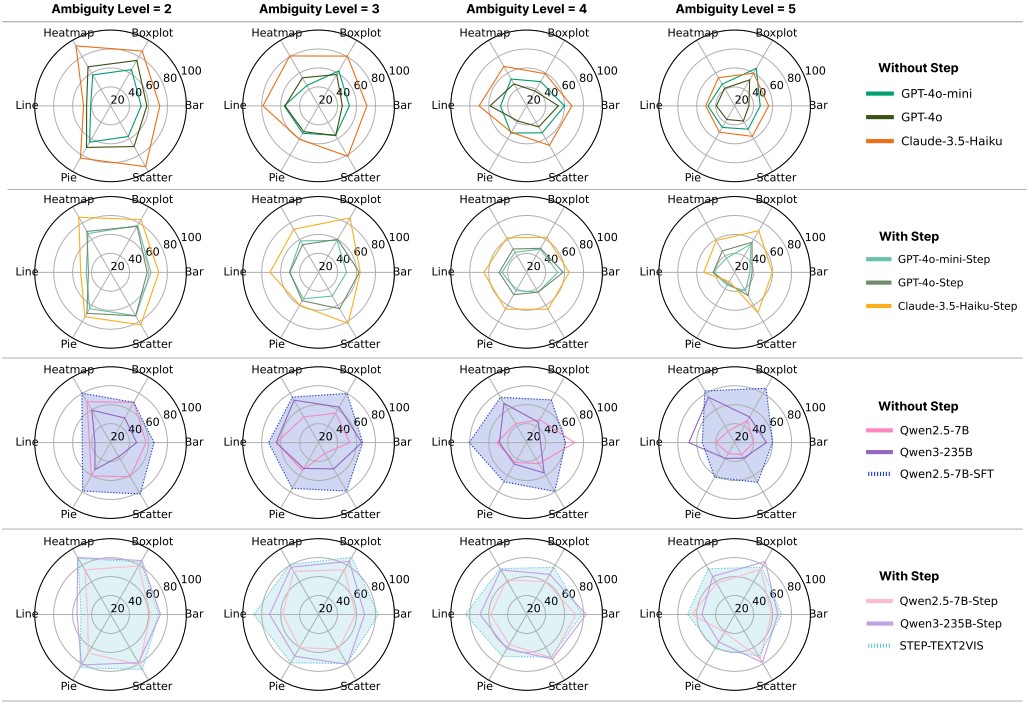

Figure 4: F1 scores across different models and ambiguity levels. The figure is organized as a 4×4 grid where columns represent increasing ambiguity levels, and rows represent different model groups. The first two rows display GPT, Claude model families with prompting-based methods. The last two rows display Qwen model families of prompting-based, supervised and preference learning methods, including our proposed Step-Text2Vis in the bottom row. Each radar chart displays F1@5 scores across six chart types, where larger polygons indicate better performance.

the "-Step" variants show notable improvements in F1 scores compared to their direct prompting counterparts. This validates our hypothesis that decomposing complex vis reasoning into explicit steps helps resolve ambiguity more effectively. Fine-tuning on nvBench 2.0 substantially improves recall at higher $K$ values. Qwen2.5-7B-SFT achieves 75.79% F1@3 and 75.30% F1@5, significantly outperforming prompt-based methods of similar model size and those from the same model family, indicating superior coverage of the valid vis space. However, this approach sacrifices some precision compared to prompting-based methods. Finally, our preference-optimized Step-Text2Vis achieves the best balance between precision and recall. At $K = 1$, it maintains exceptional precision (99.20%) while improving recall over all baselines. At $K = 3$ and $K = 5$, it achieves substantial gains in recall without significant precision degradation, demonstrating the effectiveness of step-wise preference optimization for ambiguous *Text2VIS* tasks.

**Performance Analysis Across Chart Types.** Figure 4 presents a radar chart of F1@5 scores for different methods across various chart types and ambiguity levels. We have the following observations.

First, Step-Text2Vis consistently outperforms other models across most chart types and ambiguity levels. These results demonstrate that the step-wise reasoning approach significantly enhances performance on ambiguous *Text2VIS* tasks. Second, models with step-wise reasoning (those with "-Step" suffix) generally outperform direct prompting models, confirming the effectiveness of breaking complex vis reasoning into explicit steps.

- **The Impact of Chart Types.** The experimental results reveal that different chart types exhibit varying challenges for models. Boxplot and Scatter charts generally achieve higher F1 scores, indicating they are easier for models to handle. In contrast, Pie charts perform worse at higher ambiguity levels, while Line charts consistently show lower accuracy across all ambiguity levels, with F1 scores only around 40% to 51%, even at lower ambiguity levels. These findings suggest that certain chart types pose greater challenges for model interpretation and generation.

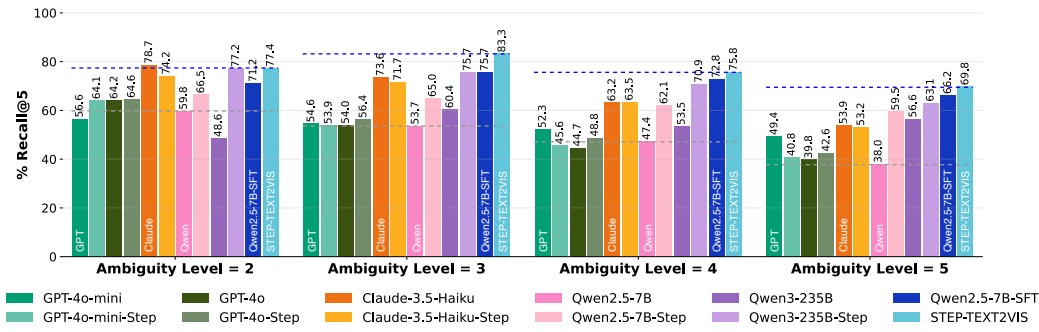

Figure 5: Recall@5 across different models and ambiguity levels. The blue dashed horizontal line indicates the performance of our proposed Step-Text2Vis method, while the grey dashed horizontal line represents Qwen2.5-7B-SFT, which serves as the base model for our approach.

- **The Impact of Ambiguity Levels.** The data shows a clear degradation in performance as the ambiguity level increases: At ambiguity level 2, most models maintain relatively high F1 scores (60-80%). By ambiguity level 5, even the best performing models struggle to maintain the same level of performance. For instance, Claude-3.5-Haiku and Claude-3.5-Haiku-Step maintain over 80% F1 scores on Heatmap, Boxplot, Pie, and Scatter charts at ambiguity level 2, yet decline to below 60% for these chart types at ambiguity level 5. Moreover, Qwen2.5-7B-Step achieving 41.55% and Step-Text2Vis achieving 61% F1 score for pie charts at this highest ambiguity level. This pattern confirms the inherent challenge of handling highly ambiguous text queries.
- **Step-wise reasoning enhances performance but alters strengths for certain models.** Most prompting-based models exhibit improvements in performance when utilizing step-wise reasoning, while still maintaining their original strengths across chart types and ambiguity levels. However, for Qwen2.5-7B, the introduction of step-wise reasoning leads to notable shifts in its area of expertise. Specifically, Qwen2.5-7B-Step demonstrates significant improvements in Boxplot (74.47%) and Heatmap (72.59%) generation at Ambiguity Level 3—an enhancement that was not prominently observed in the base Qwen2.5-7B model. This suggests that step-wise reasoning not only improves overall performance but also reshapes the model's proficiency across different tasks.

**Performance Analysis on Ambiguity Resolution Ability.** Figure 5 illustrates the Recall@5 metric, measuring each model's ability to generate valid VIS from text queries with varying ambiguity levels. Our Step-Text2Vis shows superior recall performance across all ambiguity levels. At ambiguity level 3, it achieves 83.3% recall, significantly outperforming other models. Further analysis across ambiguity levels reveals the following insights:

- **Step-wise reasoning significantly enhances performance.** Models implementing step-by-step reasoning methodologies consistently demonstrate superior performance compared to their non-stepwise counterparts. For example, Qwen2.5-7B-Step exhibits markedly improved performance metrics relative to the base Qwen2.5-7B implementation.
- **Inverse correlation between performance and ambiguity.** The experimental results indicate a consistent negative correlation between recall performance and ambiguity level for the majority of models evaluated. This trend confirms the inherently increasing complexity of VIS generation as the ambiguity level intensifies.
- **Maximum performance differentiation occurs at intermediate levels of ambiguity.** The performance delta between the evaluated models reaches its maximum when AL equals 3 or 4, suggesting that these intermediate levels provide optimal conditions for discriminating between different model capabilities.
- **Fine-tuning methods yield robust performance under increasing ambiguity.** While performance degradation is observed across all models as ambiguity increases, models employing Supervised Fine-Tuning and Preference Learning methodologies maintain superior performance characteristics at elevated ambiguity levels. Notably, the performance differential between Step-Text2Vis and alternative approaches expands proportionally with increasing ambiguity.

**Key Implications for Ambiguous *Text2VIS* Systems.** Our experimental results reveal several important implications for the design of *Text2VIS* systems that can effectively handle ambiguity. First, the performance improvements achieved through step-wise reasoning highlight the importance of decomposing complex tasks into interpretable steps, similar to how humans reason through ambiguous text queries, rather than relying on direct translation. Second, the observed performance variations across different chart types and ambiguity levels suggest that future *Text2VIS* systems should adaptively select reasoning strategies based on both the queries' characteristics and the target vis type. Third, the superior performance of Step-Text2Vis, particularly at higher ambiguity levels, demonstrates that preference-optimized models can learn to effectively balance precision and recall, maintaining high accuracy while capturing the full range of valid interpretations. These findings point toward a paradigm shift in Step-Text2Vis development: from single-output systems toward multi-interpretation frameworks that explicitly model and resolve ambiguity through structured reasoning processes.

## 5    Related Work

***Text2VIS* benchmarks.** As the predecessor of nvBench 2.0, nvBench 1.0 [24] is a commonly used *Text2VIS* benchmark that leverages the semantic alignment between queries and Visualization Query Language to construct datasets. Building on this, benchmarks like Dial-NVBench [27] introduce multi-turn dialogues, and VisEval [37] further expand *Text2VIS* evaluation. However, they primarily focus on well-specified queries with a single correct visualization. While datasets like ChartGPT [38], Text2Analysis [39], QUDA [26], and NLV [25] include ambiguous queries, they lack explicit ambiguity type definitions and comprehensive valid chart sets. In contrast, we introduce nvBench 2.0, the first ambiguity-aware *Text2VIS* benchmark designed to address this gap.

**LLMs for Data Synthesis.** LLMs have shown promise in data synthesis across various domains, enhancing data diversity and model generalization [40–52]. This includes the *Text2VIS* domain, where VL2NL [23] utilizes LLMs to generate descriptions from vis. Our approach shares a "reverse engineering" philosophy with ScienceBenchmark [53], and generates queries from ambiguity-aware vis tree to capture ambiguity in *Text2VIS* pairs, leveraging the structured nature of vis to define ambiguity types. Furthermore, we leverage LLMs to generate multi-step reasoning data, following the effectiveness demonstrated in works like Hunter et al. [54] and Step-DPO [31], to improve model reasoning and interpretability in *Text2VIS* tasks.

A more detailed discussion of related work can be found in Section A in the Appendix.

## 6    Conclusion

In this work, we introduced nvBench 2.0, the first benchmark designed for evaluating *Text2VIS* systems in scenarios involving ambiguous user queries. nvBench 2.0 was generated through a controlled ambiguity-injection pipeline, guaranteeing valid and interpretable results while offering step-wise disambiguation reasoning paths. By using nvBench 2.0, we offer a robust framework to assess *Text2VIS* systems' ability to handle ambiguities that arise in real-world applications.

We also proposed Step-Text2Vis, an LLM-based *Text2VIS* model trained on nvBench 2.0, which significantly improves *Text2VIS* performance in ambiguous scenarios by applying step-wise preference optimization. Our experimental results demonstrate that Step-Text2Vis outperforms all existing baselines, establishing a new state-of-the-art for handling ambiguity in *Text2VIS* tasks.

## Acknowledgments and Disclosure of Funding

This paper was supported by Young Talent Support Project of Guangzhou Association for Science and Technology (QT-2025-001); the NSF of China (62402409); Guangdong Basic and Applied Basic Research Foundation (2023A1515110545); Guangzhou Basic and Applied Basic Research Foundation (2025A04J3935); Guangzhou-HKUST(GZ) Joint Funding Program (2025A03J3714); and Guangdong Provincial Project (2023CX10X008).

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

# Technical Appendices and Supplementary Material

- Section A: Detailed Related Work
- Section B: More Details of Synthetic Pipeline
- Section C: More Details of nvBench 2.0
- Section D: More Details of Experimental Setups
- Section E: More Details of Error Analysis
- Section F: Limitations
- Section G: Ethic Statement

# A    Detailed Related Work

## A.1    *Text2VIS* Benchmarks

*Text2VIS* benchmarks play a crucial role in evaluating the performance of *Text2VIS* systems [55]. As the predecessor of nvBench 2.0, nvBench 1.0 [24] is a commonly used *Text2VIS* benchmark, constructs datasets by leveraging the semantic alignment between SQL and visualization query language, which is a SQL-like specification that defines the visualization structure and details the data transformation processes. It employs template-based structures to systematically translate VQL into NL. This structured approach facilitates end-to-end model training by enhancing the clarity of both inputs and outputs [36, 56, 57, 7, 58, 59]. Building on nvBench 1.0 [24], Dial-NVBench [27] introduces multi-turn dialogues, allowing models to capture user intent through iterative interactions. VisEval [37] further refines nvBench by filtering out ambiguous, irrational, duplicated, and incorrect queries using a three-step selection process (rule-based, LLM-based, and human-based), and offers an automated evaluation framework covering validity, legality, and readability. However, all three benchmarks [24, 27, 37] remain focused on well-specified queries that map directly to a single correct visualization, without explicitly addressing ambiguity in user intent.

To explore ambiguous and under-specified query formulations, ChartGPT [38] extends nvBench by prompting GPT-3 to generate more abstract and natural utterances compared to the original ones. Similarly, while some other *Text2VIS* datasets include ambiguous queries [39, 26, 25], they do not explicitly define ambiguity types and provide a complete set of valid chart results. Beyond the realm of *Text2VIS*, ambiguity has also been explored in *Text2SQL* benchmarks, where studies have considered data selection and computation ambiguity [18, 60], but they do not address ambiguity in the visualization space. While some *Text2VIS* systems have attempted to address ambiguity by detecting it [15, 61] or inferring underspecified queries [62], they lack a benchmark for systematic evaluation.

To fill this gap, we propose nvBench 2.0, the first ambiguity-aware *Text2VIS* benchmark, which provides ambiguous user queries and supports one-to-many mappings with multiple valid visualizations. By doing so, it enables a more comprehensive evaluation of *Text2VIS* systems in real-world scenarios.

## A.2    LLMs for Data Synthesis

Recently, the use of LLMs for data synthesis or data augmentation has become increasingly prevalent. Many studies leverage LLM-generated data for training models [40–45, 63], as well as for evaluating the performance of other trained models [49]. In the NLP domain, researchers have utilized LLMs to generate synthetic data for tasks like text classification [50–52]. These works showcase that LLM-generated data can enhance data diversity, thereby improving model generalization and robustness. Building on this, VL2NL [23] extends LLMs to *Text2VIS* domain, generating natural language descriptions (*e.g.,* L1 and L2 captions, and user commands) from Vega-Lite specifications. Similarly, the application of LLMs for tabular data or database-related tasks has gained attraction. Common approaches for generating *Text2SQL* or table question answering datasets often involve generating TEXT queries first, followed by SQL generation [18, 60]. ScienceBenchmark [53] takes a reverse approach by starting with seed SQL queries, then generating new SQL queries from the domain schema, and translating them into natural language queries using fine-tuned LLMs. We follow this reverse construction philosophy in developing nvBench 2.0. Specifically, we begin by extracting VQL from seed charts and then use LLMs to reverse engineer the corresponding text descriptions. The advantage of this approach is that VQL clearly defines each step and the ambiguity types involved, allowing us to better capture one-to-many (TEXT, VIS) pairs.

By leveraging LLMs to generate multi-step reasoning data, the performance of models on long-chain and complex reasoning tasks can be further improved. As demonstrated by Hunter et al. [54], process supervision via multi-step reasoning significantly enhances model reliability on tasks such as mathematical problem-solving. Similarly, Step-DPO [31] shows that generating step-wise reasoning data enables models to better capture intermediate steps, resulting in improved accuracy. Following this approach, we also generate multi-step reasoning data for tasks in the *Text2VIS* domain, where each step of the reasoning process is explicitly defined, contributing to more accurate and interpretable model predictions.

## B   More Details of Synthetic Pipeline

### B.1   Step 1: Ambiguity-aware VIS Tree Synthesis

Our ambiguity-aware visualization tree synthesis forms the foundation for synthesizing ambiguous *Text2VIS* data. As shown in Figure 6, this process injects ambiguities into a seed visualization.

**Transforming the Seed Visualization into a Tree Abstraction.** Given a data table $D$ and a seed visualization $v$ (*e.g.,* (Figure 6-①), we first convert the $v$—along with its underlying query—into an Abstract Syntax Tree (AST), which we refer to as the seed visualization tree $T$ (*e.g.,* Figure 6-②). The grammar of AST is based on the predecessor work, nvBench [24]. This tree explicitly encodes all design decisions made in creating $v$ and is formally defined as:

$$v \mapsto T = \{\mathbf{A} \mid \mathbf{A} = [a_1, a_2, \ldots, a_t]\} \tag{2}$$

Here, each node $a_i$ represents a construction action for a visualization component as a tuple $(\tau, op, params)$, where:

- $\tau \in \{\texttt{explicit}, \texttt{ambiguous}, \texttt{implicit}\}$ denotes the ambiguity type of the action node;
- $op$ specifies the operation (*e.g.,* data selection, chart type selection, channel mapping, data transformation selection, etc.);
- $params$ contains the specific parameters for the operations.

**Controlled Ambiguity Injection.**

We then transform $T$ into an ambiguity-aware tree $T'$ through three operations:

- Injecting ambiguous nodes : We add nodes that represent components with multiple valid interpretations. For example, replacing "Local Gross" with an ambiguous choice between "Local Gross" and "World Gross".
- Adding implicit nodes : We include nodes for components not explicitly specified but required for visualization completion. For example, adding a node for the "COLOR" encoding channel.
- Modifying explicit nodes : We adjust certain explicit nodes to account for potential ambiguities. For example, changing a "Mark" node initially set as "Bar" into an ambiguous choice among various mark types or requiring inference from analytic tasks.

By applying these steps, the resulting ambiguity-aware tree $T'$ captures the full range of possible interpretations for the seed visualization. For example, as shown in Figure 6-③, this tree contains some new nodes such as:

A1 : (ambiguous, data_column, {field:[Local_Gross, World_Gross]})

A2 : (explicit, task, {value:[Trend]})

A3 : (implicit, data_value, {value:[Comedy, Action]})

**Ambiguity Metadata Generation for Ambiguity Injection.**

To enable precise ambiguity injection in data tables, we propose a systematic metadata generation process that integrates structured knowledge bases with large language models (LLMs). This process

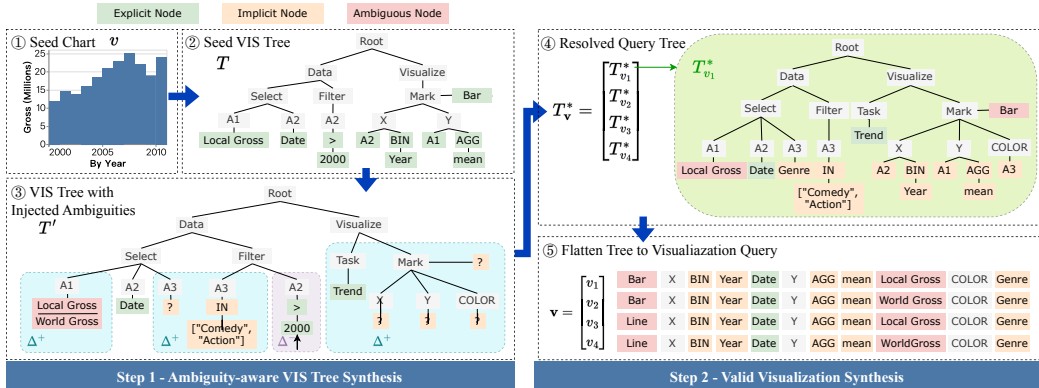

Figure 6: Injecting ambiguities into a seed visualization.

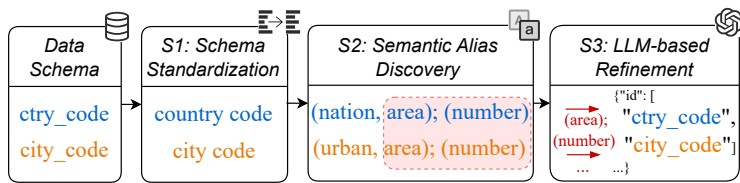

Figure 7: Ambiguity metadata generation workflow.

identifies and categorizes potential semantic ambiguities in table schemas, producing metadata that guides the construction of ambiguity-aware visualization trees. Each node in the visualization tree $T'$ is labeled as ambiguous, implicit, or explicit based on the metadata, ensuring visualizations reflect multiple valid query interpretations. The process comprises three key stages: schema standardization, semantic alias discovery, and LLM-based refinement.

*Stage 1: Schema Standardization:* The first step involves standardizing the original data schema by refining or expanding column names. Abbreviated or domain-specific terms are transformed into more descriptive, conventional labels. For example, a column labeled `ctry_code` is standardized to `country code`. Such standardization forms a clearer basis for subsequent ambiguity analysis.

*Stage 2: Semantic Alias Discovery:* After standardizing the schema, we leverage ConceptNet [64] to identify potential semantic aliases for each column name. ConceptNet's multilingual knowledge graph provides synonyms, hypernyms, and other semantically related terms, helping detect conceptual overlaps. We flag pairs of columns with similar meanings or concept overlap as potential sources of ambiguity. For example, `country code` and `city code` may both have meanings related to `area number`, introducing possible confusion in user queries.

*Stage 3: LLM-Based Refinement:* We refine the flagged ambiguous column pairs using GPT-4o-mini with a chain-of-thought (CoT) prompting strategy. The model analyzes the original column names, their standardized forms (Stage 1), and the ConceptNet-derived aliases and ambiguity flags (Stage 2). It then generates a final, validated set of ambiguous pairs, which is formatted into a JSON metadata file. For example, as shown in Figure 7, one of the identified ambiguous pairs is `ctry_code` and `city_code` due to their similar word aliases. This process supports ambiguity-aware visualization generation and step-wise reasoning.

By combining these stages, we generate the necessary metadata to guide the construction of ambiguity-aware visualization trees, ensuring that each node is accurately marked as explicit, ambiguous, or implicit, thus enabling the synthesis of visualizations that reflect multiple valid interpretations of the query.

## B.2 Step 2: Valid Visualization Synthesis

Once we have an ambiguity-aware visualization tree $T'$, the next stage is to generate a set of valid visualizations $\mathbf{v} = \{v_1, v_2, \ldots, v_k\}$. Each visualization $v_i$ represents one possible resolution of the ambiguities present in $T'$ (see Figure 6-③). In this step, we define a resolution function $\mathcal{R}$

to systematically clarifies ambiguous and implicit nodes, transforming $T'$ into a set of resolved trees $\{T^*_{v_1}, \ldots, T^*_{v_k}\}$ (see Figure 6-④). Each resolved tree $T^*_{v_i}$ is then "flattened" into a concrete visualization query $v_i$ (see Figure 6-⑤).

**Task Description.** Recap that a partially ambiguous visualization tree $T'$ may contain:

- *Ambiguous nodes*: Multiple valid interpretations (*e.g.,* which column to use for "gross").
- *Implicit nodes*: Necessary but unspecified details (*e.g.,* binning a date field by year).
- *Explicit nodes*: Directly specified components (*e.g.,* "bar" mark).

To produce valid visualizations, these ambiguous and implicit nodes must be resolved in a manner consistent with established visualization grammar rules (*e.g.,* requiring temporal fields to be binned). Formally, we define:

$$\mathcal{R}(T') \rightarrow \{T^*_{v_1}, T^*_{v_2}, \ldots, T^*_{v_k}\} \tag{3}$$

where each $T^*_{v_i}$ is a resolved tree that has no remaining ambiguity or unspecified details. The flattening process then converts each $T^*_{v_i}$ into a finalized visualization specification $v_i$. This yields the complete set of valid visualizations: $\mathbf{v} = \{v_1, v_2, \ldots, v_k\}$.

In the following sections, we describe how an Answer Set Programming (**ASP**) solver [30] is used to implement the resolution function $\mathcal{R}$ while ensuring that each resolved visualization adheres to the necessary grammar constraints.

**ASP Solver Objective.** ASP is a declarative constraint programming paradigm well-suited for knowledge representation and reasoning [30, 65, 66]. Encoding the ambiguity resolution process and grammar rules as logical constraints has the following benefits:

- *Completeness*: The solver can enumerate all stable models (*i.e.,* all possible ways to resolve ambiguous or implicit nodes) that satisfy the visualization grammar.
- *Correctness*: Only solutions that meet mandatory constraints (*e.g.,* "temporal fields must be binned") are considered valid.
- *Diversity*: Each output corresponds to a distinct interpretation of the query, ensuring coverage of all plausible visualizations.

The number of resulting visualizations, $k = |\mathbf{v}|$, represents the **ambiguity level**—how many distinct interpretations the solver deems valid for the given $T'$. After obtaining these solutions, we can filter or select a subset based on a target ambiguity level $k$, ensuring that each retained visualization differs from the others.

**ASP Syntax Overview.** ASP is built on a logical foundation with several key syntactic constructs [30]. The fundamental unit in ASP is a rule of the form: `Head :- Body.`, which states that the head is true if all literals in the body are satisfied. For example, the rule: `light_on :- power_available, switch_flipped.` expresses that the light will be on if both power is available and the switch is flipped.

Some special cases include:

- *Facts*: Rules without a body represent unconditional truths. For example, `power_available.` asserts that power is available.
- *Integrity Constraints*: Rules without a head prohibit certain combinations of conditions. For example, the constraint: `:- not power_available, light_on.` ensures that the light cannot be on when power is not available.

An ASP program consists of a collection of rules, facts, and constraints that collectively define a search space. The ASP solver then computes all stable models (*i.e.,* answer sets) that satisfy these conditions. Each stable model represents a valid system state or, in our context, a valid resolution of the ambiguous visualization tree.

For example, consider a simple lighting system modeled with:

- *Rule:* `light_on :- power_available, switch_flipped.`

- *Fact:* `power_available.`, `switch_flipped.`

Given these statements, the ASP solver determines the unique answer set containing `light_on`, as all conditions in the rule body are satisfied. If we instead had `not switch_flipped.`, the solver would exclude `light_on` from the answer set.

By exhaustively computing all stable models that meet the specified constraints, the ASP solver identifies all valid visualization configurations implied by our ambiguity-aware visualization tree. This systematic resolution is key to generating a complete set of valid visualizations from an ambiguous query.

**ASP Rules for Resolving Ambiguity-aware Visualization Tree.** We formalize the visualization design space using ASP by converting each node in the ambiguity-aware visualization tree $T'$ into ASP rules. As defined in Section B.1, each node in the visualization tree is represented as a tuple $(type, operation, parameters)$, which is mapped into ASP entities, (*e.g.,* like `entity(E, _, _).`) and their associated attributes (*e.g.,* like `attribute(A, _, _).`.

*Rules for Explicit Nodes.* Nodes that directly specify a visualization component are encoded as entities with fully defined attributes. For example, a node indicating a specific mark selection, such as a bar chart, is encoded in ASP as:

- `entity(mark, parent_id, mark_id).`
- `attribute((mark, type), mark_id, bar).`

These rules explicitly assert that the mark type is"bar".

*Rules for Ambiguous Nodes.* Nodes that allow multiple valid interpretations are encoded using ASP choice rules. For example, if an encoding node can correspond to either "temp_max" or "temp_min", we encode this ambiguity as follows:

- `1 { attribute((encoding, field), e_id, temp_max); attribute((encoding, field), e_id, temp_min) }.` ensures at least one option should be selected.
- An accompanying integrity constraint ensures that only one of the two options is selected: `:- at tribute((encoding, field), e_id, temp_max), attribute((encoding, field), e_id, temp_min).`

This formulation forces the solver to choose exactly one interpretation for each ambiguous node.

*Rules for Implicit Nodes.* Implicit nodes represent necessary components that are not explicitly specified in the query. These nodes are encoded using placeholder attributes to indicate that the value is not determined. For example, a mark node with an unspecified chart type is represented as:

- `entity(mark, parent_id, mark_id).`
- `attribute((mark, type), mark_id, _).`

This indicates the mark exists, but its type is undetermined.

To capture the complete visualization design space, we also encode comprehensive design knowledge as ASP rules [67, 65, 66], which fall into three categories:

*Definition Rules for Visualization.* Declarative statements that establish foundational visualization elements, such as available chart types or encoding channels. For example, `domain((mark, type),(point; bar; pie)).` defines that the mark type for a chart can be point, bar, or pie.

*Hard Constraints for Visualization.* Mandatory conditions that any valid visualization must satisfy. For example, the constraint `violation(no_encodings) :- entity(mark,_,M), not entity(encoding,M,_).` ensures that every mark has at least one visual encoding channel.

*Choice Rules for Visualization.* Rules that govern the selection among multiple options when constructing a visualization. For example `0 { attribute((encoding, field), E, N): domain((field, name), N) } 1 :- entity(encoding,_, E).` ensures that each encoding is associated with at most one field.

**Applying ASP Solver to Reason Valid Visualization.** By encoding the ambiguity-aware visualization tree structure and design principles as ASP rules, we create a powerful mechanism to resolve ambiguities. The ASP solver explores all possible resolutions for ambiguous nodes, ensuring that only solutions adhering to the visualization grammar constraints are accepted. This results in a diverse set of valid visualizations, with variations in chart type, encoding mappings, and data transformations, while staying true to the original ambiguous query.

### B.3 Step 3: Ambiguous Text Query Synthesis

As shown in Figure 2 (c), this step runs in parallel with the valid visualization synthesis described in Section B.2. Building on the ambiguity-aware visualization tree $T'$, this step aims to synthesize a corresponding ambiguous natural language query $q$.

**Task Description.** Given the input ambiguous visualization tree $T' = \{\mathbf{A} \mid \mathbf{A} = [a_1, a_2, \ldots, a_h]\}$, the corresponding natural language query $q$ is generated using the mapping function $\mathcal{M}$:

$$Q = \mathcal{M}(T') = [\mathcal{M}(a_1), \mathcal{M}(a_2), \ldots, \mathcal{M}(a_h)] \tag{4}$$

where the tuple of each visualization construction action $a_i$ in $T'$ is mapped to a corresponding natural language expression $\mathcal{M}(a_i)$.

For a given $T'$, its corresponding $q$ must satisfy the following conditions to ensure correctness:

- *Completeness*: Ensure that all actions in the original $T'$ are covered in the generated $q$:

$$\forall a_i \in T', \exists \mathcal{M}(a_i) \in Q \tag{5}$$

- *Type Preservation*: $q$ must preserve the ambiguity types of the original action nodes:

$$\tau(\mathcal{M}(a_i)) = \tau(a_i), \quad \forall a_i \in T' \tag{6}$$

  where $\tau(a_i)$ is the ambiguity type of action node $a_i$.
- *Boundedness*: $q$ should not introduce any actions outside of $T'$:

$$\forall \text{ expression } e \in Q, \exists a_i \in T' : e = \mathcal{M}(a_i) \tag{7}$$

**Solution Overview.** We leverage an LLM-based `Text Query Generator` to integrate the ambiguities introduced in $T'$ into a single and coherent query $q$, ensuring that the generated query faithfully reflects all the intended ambiguous components. Finally, an `Text Query Verifier` is employed to validate that $q$ accurately captures the ambiguity without introducing any extraneous semantics. This two-step process—generation followed by verification—ensures that the final query remains consistent with the design decisions encoded in $T'$ while meeting the criteria of completeness, type preservation, and boundedness.

**Text Query Diversity in Generation.** NLV Corpus [25] defines several distinct categories of natural language utterances—question, command, query, and other. Since "query" somewhat overlaps with other styles, we focus on three main types: question, command, and caption, each representing a distinct style of user input:

- *Question*: Typically begins with a question word (*e.g.,* "What", "How much", "How many", etc.).
- *Command*: Usually an imperative sentence (*e.g.,* "Show a bar chart of sales by region").
- *Caption*: Includes non-standard phrases, incomplete sentences, or informal text conveying user intent, often brief (*e.g.,* "SUM (Sales) vs Date" or "budget over time").

To ensure diversity of the generated queries, we provide specific Text Query styles and corresponding example queries as input to the language model. These examples are randomly sampled from a large corpus to ensure variability.

**Text Query Generator.** To systematically align the structured visualization tree with diverse natural language expressions, we define explicit input-output mappings. The input to the LLM (GPT-4o-mini-turbo) delivers essential context, including data schema, sample data, action sequences, and style requirements. This aims to ensure that the output text query: maintains linguistic grounding for all

Table 4: Chart types, visual channels, and analytic tasks with compatible data types: $C$=Categorical, $Q$=Quantitative, $T$=Temporal, $\emptyset$=N/A.

| Chart Type | Encoding Channel x\|y\|color\|size\|theta | Analytic Task |
|:---:|:---:|:---:|
| Bar | $\{C,Q,T\}\|Q\|C\|\emptyset\|\emptyset$ | Trend, Distribution |
| Line | $\{C,Q,T\}\|Q\|C\|\emptyset\|\emptyset$ | Trend, Distribution |
| Pie | $\emptyset\|\emptyset\|C\|\emptyset\|Q$ | Distribution |
| Scatter | $Q\|Q\|C\|Q\|\emptyset$ | Correlation |
| Heatmap | $\{C,Q,T\}\|\{C,Q\}\|Q\|\emptyset\|\emptyset$ | Correlation |
| Boxplot | $\{C\}\|Q\|C\|\emptyset\|\emptyset$ | Distribution |

actions (5), preserves ambiguity types during translation (6), and avoids introducing any extraneous semantics (7). The complete prompt format is in Table 7.

**Text Query Verifier.** As indicated by recent studies [68, 69], LLMs outputs still require verification, particularly concerning *boundedness* (7). The verification can be performed by LLMs or human evaluators. In our preliminary experiments, we found that LLM-based verification is sufficient to achieve an accuracy of 99%. Thus, we designed the following prompt for LLM verification as shown in Figure 9

If $L_1$ fully covers all nodes in $T$ while $L_2$ remains empty, the $q$ is considered valid and added to the dataset. Otherwise, $q$ is classified as invalid, and it would be regenerated by the **Text Query Generator**. This approach checks for *completeness* (5), *type preservation* (6), and *boundedness* (7). If the verification fails, the system can regenerate the query or suggest corrections.

## B.4 Step 4: Ambiguity-resolved Reasoning Path

Based on the previous discussion, we have reformulated the *Text2VIS* problem from a direct mapping $q \rightarrow \mathbf{v} = \{v_1, \ldots v_k\}$ to a structured process $q \rightarrow T' \rightarrow T_{\mathbf{v}}^* \rightarrow \mathbf{v}$. To mimic human-like reasoning workflow for ambiguity resolution, we propose decomposing the ambiguity-aware visualization generation process into a sequential reasoning path with five distinct steps, as illustrated in Figure 1:

$$q \xrightarrow{\phi_1} S_1 \xrightarrow{\phi_2} S_2 \xrightarrow{\phi_3} S_3 \xrightarrow{\phi_4} S_4 \xrightarrow{\phi_5} \mathbf{v} \tag{8}$$

where each $\phi_i$ represents a reasoning function and each $S_i$ represents the intermediate state after applying the corresponding reasoning function.

**Step-①: Data Selection Reasoning.** The first step parses the natural language query $q$ into data components from the data table:

$$\phi_1(q) \rightarrow S_1 = \{a_1^c, a_2^c, \ldots, a_m^c\} \tag{9}$$

where each $a_i^c$ represents a data component selection action, including column selection, value selection, and filter condition specification. The outcomes of this step correspond to the SELECT and FILTER nodes in the visualization tree (see Figure 6).

**Step-②: Chart Type Reasoning.** The second step determines appropriate visualization mark types based on the analytic task:

$$\phi_2(S_1, q) \rightarrow S_2 = S_1 \cup \{a_1^v, a_2^v, \ldots, a_n^v\} \tag{10}$$

where each $a_i^v$ represents a visualization design action, including analytic task identification and chart type selection. As shown in Table 4, existing visualization design principles [70**?**, 71] can establish a mapping relationship between tasks and chart types [70**?**, 72]. When the text query $q$ does not

explicitly specify a chart type, the identified task can guide inference, though this may introduce ambiguity as multiple chart types may be suitable for a given task. In addition, certain tasks influence encoding channel selection in the next step.

**Step-③: Channel Mapping Reasoning.** The third step establishes the mappings between data components and encoding channels:

$$\phi_3(S_2) \to S_3 = S_2 \cup \{a_1^m, a_2^m, \dots, a_p^m\} \tag{11}$$

where each $a_i^m$ represents a channel mapping action, such as assigning data columns to encoding channels like X, Y, color, or size. This step ensures that data columns are mapped appropriately, aligning with visualization design principles, where some mapping relationships are shown in Table 4.

**Step-④: Data Transformation Reasoning.** The fourth step specifies necessary data transformations based on the channel mappings:

$$\phi_4(S_3) \to S_4 = S_3 \cup \{a_1^t, a_2^t, \dots, a_r^t\} \tag{12}$$

where each $a_i^t$ represents a data transformation action, including aggregation, binning, sorting, and filtering operations, these transformations prepare the data to be properly visualized according to the selected chart type and channel mappings.

**Step-⑤: Visualization Synthesis Reasoning.** The final step is to integrate all reasoning steps to generate a set of valid visualizations:

$$\phi_5(S_4) \to \mathbf{v} = \{v_1, v_2, \dots, v_k\} \tag{13}$$

where each $v_i$ represents a valid visualization specification. This process can produce multiple valid visualizations that address different aspects of the ambiguity in the original query (see Figure 1).

The four reasoning steps (Step-① to Step-④) outlined in the ambiguity-resolved reasoning path are not strictly bound by a fixed sequence and can be executed in any order, provided all steps are completed before the final visualization synthesis (Step-⑤). This flexibility arises because each step addresses a distinct aspect of the visualization process—data selection, chart type reasoning, channel mapping, and data transformation—and their interdependencies are managed through the shared ambiguity-aware visualization tree $T'$. The exact order may vary depending on the text query; for example, if the query lacks any clues about the chart type, chart type reasoning may occur last, after all selected data and possible channel mappings have been considered.

This structured reasoning process systematically addresses ambiguity at each step while adhering to visualization design principles. Each step builds upon prior decisions, progressively refining the visualization specifications to account for multiple valid interpretations of the original text query.

Formally, the complete reasoning path can be expressed as the composition of the step-wise reasoning functions:

$$\mathcal{F}(q, D) = (\phi_5 \circ \phi_4 \circ \phi_3 \circ \phi_2 \circ \phi_1)(q, D) \to \mathbf{v} \tag{14}$$

This decomposition simplifies the ambiguity-aware *Text2VIS* process, breaking down complex reasoning into steps that better align with LLMs' strengths in natural language understanding and generation. Techniques like chain-of-thought prompting or step-wise direct preference optimization (step-DPO) [31, 54] can further improve LLM performance.

Finally, as shown in Figure 2 (d), the LLM-based step-wise reasoning generator takes the text query $q$, the generated unambiguous visualization $v$, and the ambiguous visualization tree $T'$ as input. It then performs reverse reasoning for each step (14), generating text-based reasoning descriptions. For example, when resolving chart type ambiguity in Figure 1, the LLM reasons, "Since this query requests a trend analysis over time, either bar charts or line charts would be appropriate, as both effectively represent temporal patterns in the data" for Step-②. The complete prompt format is in Table 9.

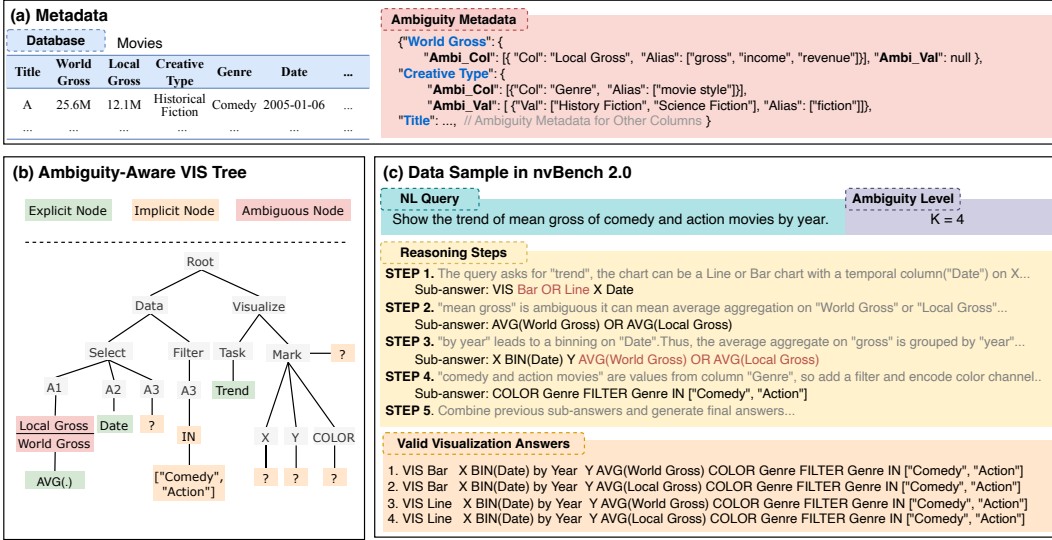

Figure 8: An example in nvBench 2.0.

# C More Details of nvBench 2.0

## C.1 Detailed Example in nvBench 2.0

Figure 8 illustrates an example sample in the nvBench 2.0, showcasing how the data is stored and the information it contains. Figure 8 (a) Presents the data schema for the "Movies" database, incorporating ambiguity metadata that highlights potential ambiguities, such as column aliases (e.g., "World Gross" and "Local Gross" both linked to "gross") and value aliases (e.g., "History Fiction" and "Science Fiction" both linked to "fiction").

Figure 8 (b) Displays the Ambiguity-Aware VIS Tree, depicting the hierarchical structure of ambiguous user intent, with explicit nodes shown in green, implicit nodes in yellow, and ambiguous nodes in red, revealing the underlying ambiguity intents for the data sample in (c).

Figure 8 (c) represents a data sample within nvBench 2.0, beginning with the text query "Show the trend of the mean gross of comedy and action movies by year" and an ambiguity level of $K = 4$, demonstrating the number of gold answers available for this query. The sample includes reasoning steps, each with 1-2 sentences of logical reasoning and a sub-answer, culminating in the four gold answers.

## C.2 Detailed Statistics of nvBench 2.0

**Data Tables.** Figure 9 (a.1) shows that most tables in our dataset have 2–5 columns, with fewer than 50 tables having more than 8 columns. As Figure 9 (a.2) illustrates (log scale), row counts range widely, from 10–1000 rows for many tables to outliers exceeding 10,000 rows. This variety ensures that nvBench 2.0 tests system performance across both small and large datasets.

**Ambiguity Types and Levels.** An important contribution of nvBench 2.0 is the systematic introduction of controlled ambiguity levels. Figure 9 (b.1) categorizes ambiguity by type: Data Transformation (DT) ambiguities are most prevalent ($\sim$ 3,500 examples), followed by Channel Mapping (CM) ambiguities ($\sim$ 1,500 examples), with Data Selection (DS) and Chart Type Selection (CT) ambiguities represented by approximately 900 and 400 examples, respectively. As shown in Figure 9 (b.2), the majority of samples (approximately 3,500) have an ambiguity level of 2, indicating that two valid visualizations exist. The dataset also contains a substantial number of samples with ambiguity levels of 3, 4, and 5, enabling a thorough evaluation of systems under increasingly complex ambiguous scenarios. Figure 9 (c.2) and (d.2) further illustrates the relationship between ambiguity levels and two factors: chart types (c.2) and NL styles (d.2), showing comprehensive data coverage.

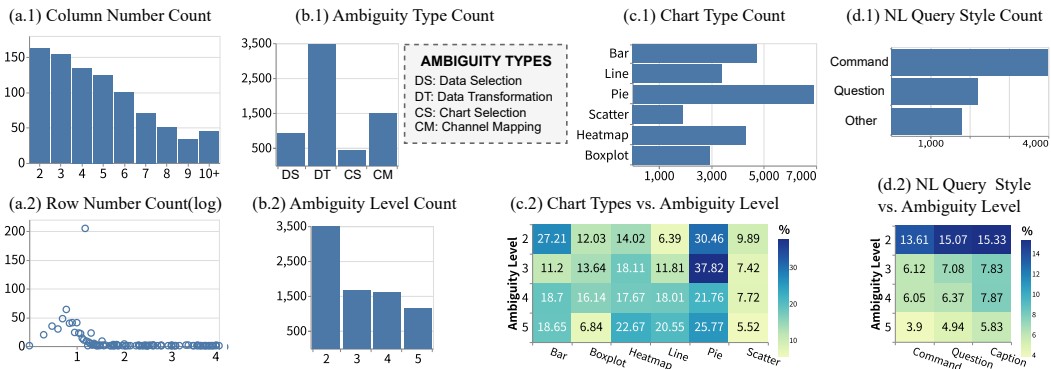

Figure 9: Detailed Statistics of nvBench 2.0.

Table 5: Distribution of natural language styles across chart types and word count statistics

| NL Style | Count by Chart Type | | | | | | Total | Word Count | | |
|---|---|---|---|---|---|---|---|---|---|---|
| | Bar | Line | Pie | Scatter | Boxplot | Heatmap | | Avg. | Max | Min |
| Command | 1368 | 922 | 1922 | 608 | 1319 | 894 | 2338 | 14.20 | 60 | 6 |
| Question | 1570 | 1084 | 2299 | 679 | 1403 | 966 | 2636 | 14.04 | 39 | 5 |
| Caption | 1779 | 1363 | 2651 | 581 | 1589 | 1079 | 2904 | 14.00 | 65 | 5 |
| Total | 4717 | 3369 | 6872 | 1868 | 4311 | 2939 | 7878 | 14.07 | 65 | 5 |

**Visualizations.** Figure 9 (c.1) shows the distribution of chart types in nvBench 2.0. Pie charts are the most common, with around 6,000 examples, followed by bar charts (∼4,000) and heatmaps (∼3,500). Additionally, line charts (∼2,800), boxplots (∼2,000), and scatter plots (∼1,500) are also well-represented, ensuring that the benchmark covers all major visualization types. This distribution reflects common visualization practices, where pie and bar charts are widely used for categorical comparisons, while the other types serve specialized analytical needs.

**Text Queries.** Figure 9 (d.1) presents the natural language query distribution. Command-based queries (*e.g.,* "Show me the sales by region") are most frequent (∼4,000). Question-based queries (*e.g.,* "What are the sales trends?") and caption-like statements (*e.g.,* "SUM (Sales) vs Date") appear in about 2,000 and 1,800 instances, respectively. Table 5 provides a detailed breakdown of NL styles across different chart types, along with word count statistics. Commands, questions, and captions are distributed across various chart types, with pie charts receiving the highest number of queries (6,872). The average word count remains consistent (∼14 words), with captions exhibiting the longest maximum length (65 words). This distribution highlights the dataset's diversity in both linguistic structure and visualization needs, ensuring that nvBench 2.0 can effectively evaluate systems' capabilities to handle diverse user interactions.

# D   More Details of Experimental setups

**Methods.** We evaluate the performance on ambiguous *Text2VIS* tasks using both prompting-based and fine-tuning-based methods with our nvBench 2.0. The primary goal is to assess the model's ability to generate diverse and semantically accurate visualizations in response to ambiguous TEXT queries.

*Prompting-based Methods.* We evaluate two prompting strategies with GPT-4o-mini, GPT-4o and Qwen2.5-7B-Instruct model:

- **Direct Prompting:** See Table 10 for complete prompt structure. The model receives structured *Data Information* and an *Text Query* as input, generating 1-5 distinct charts to cover possible interpretations of ambiguous queries.

- **Step Prompting:** See Table 11 for complete prompt structure. Models are guided to "`think step-by-step`", explicitly articulating their reasoning process before generating visualizations. Models using this approach are denoted with a "-Step" suffix.

*Supervised Fine-tuning Method.*

- **Qwen2.5-7B-SFT**: We performed supervised fine-tuning on the Qwen2.5-7B-Instruct model using the training set, enabling direct generation of multiple Vega-Lite definitions without step-wise reasoning. Training involved three epochs with a global batch size of 16, a learning rate of 2e-5, the AdamW optimizer, and a cosine learning rate scheduler with a 0.1 warmup ratio.

*Preference Learning Method.*

- **Step-Text2Vis**: We designed Step-Text2Vis to handle the ambiguity in *Text2VIS* through step-wise reasoning as detailed in Section 3. After the initial supervised fine-tuning of Qwen-2.5-7B-Instruct, we constructed a preference dataset from the nvBench 2.0 development set for Step-DPO training. This process used one epoch with a global batch size of 4, a linearly decaying learning rate from 2e-6, and the AdamW optimizer.

**Evaluation Metrics.** Detailed explanation for evaluation metrics:

- **Precision@K (P@K)**: Assesses recommendation accuracy by calculating the proportion of valid visualizations among the top-K outputs. Higher P@K indicates more trustworthy recommendations, with fewer incorrect visualizations shown to users.
- **Recall@K (R@K)**: Quantifies how completely the model covers the golden visualization space by measuring the proportion of valid visualizations successfully identified. This captures the model's ability to represent multiple valid interpretations for ambiguous queries.
- **F1@K**: Provides a balanced measure that combines precision and recall through their harmonic mean. This comprehensive metric rewards systems that achieve both high coverage of the golden answer space and high accuracy in their recommendations.

For all experiments, we report these metrics at $K \in \{1, 3, 5\}$ to evaluate performance across different recommendation set sizes.

# E   More Details of Error Analysis

## E.1   Qualitative Error Analysis in Reasoning Paths.

Figure 10 outlines frequent errors in the stepwise reasoning process for ambiguous Text2VIS tasks across four key steps:

- At the Data Selection step, errors frequently arise when models overgeneralize column references, selecting irrelevant or overly broad sets of data that do not align with the query's intent.
- During the Data Transformation step, mistakes often occur due to misinterpretation of aggregation requirements, such as applying incorrect operations like counting rows instead of summing specific values.
- At the Chart Selection step, errors commonly stem from choosing inappropriate chart types that fail to match the analytical task or data characteristics, leading to ineffective visualizations.
- In the Channel Mapping step, issues frequently emerge from incorrect assignments of data to visualization channels, such as mapping a temporal field to a color channel instead of an axis, resulting in misleading representations.

The stepwise reasoning step information provided in these examples is valuable for further ambiguous *Text2VIS* system development, as it offers insights into common reasoning pitfalls, enabling the design of more robust models that can learn from and correct these errors to improve accuracy and interpretability in handling ambiguous queries.

## Example 1: Error at Data Selection Step

**Text Query**

*"The distribution of ids shown in a pie chart."*

**Database: Stores**

| workshop_id | store_id | store_name | email_address | ... |
|---|---|---|---|---|
| 1 | A01 | Store A | a@xx.com | ... |
| 2 | B01 | Store B | b@xx.com | ... |

**Bad Reasoning Step**

**Reasoning:** The query mentions "distribution of ids" and a "pie chart." "Ids" likely refers to any ID-like column, so I'll select workshop_group_id, address_id, store_name, and store_email_address.

**Sub-Answer:** SELECT [workshop_group_id, address_id, store_name, store_email_address].

**Good Reasoning Step**

**Reasoning:** The query mentions "distribution of ids" and a "pie chart." "Ids" could refer to workshop_group_id or address_id, not store_name or store_email_address.

**Sub-Answer:** SELECT [workshop_group_id, address_id].

**Problem:** Overgeneralizes "ids" to include non-ID columns, leading to incorrect answers.

## Example 2: Error at Data Transformation Step

**Text Query**

*"Show the distribution of games played by team in a bar chart."*

**Database: Player**

| player | team | game_played | season | ... |
|---|---|---|---|---|
| Alice | Team A | 25 | 2024 | ... |
| Bob | Team B | 30 | 2024 | ... |

**Bad Reasoning Step**

**Reasoning:** The query asks for "distribution of games played by team." I'll aggregate by counting rows for each Team to show distribution.

**Sub-Answer:** SELECT COUNT(*) GROUP BY team

**Good Reasoning Step**

**Reasoning:** The query requests "distribution of games played by team." Game_Played is already a count, so I'll aggregate it (sum or average) for each Team.

**Sub-Answer:** SELECT SUM(game_played) GROUP BY team

**Problem:** Incorrectly applies a count of rows, ignoring that game_played already represents game counts, leading to player counts per team.

## Example 3: Error at Chart Selection Step

**Text Query**

*"Distribution of total sales in 2024 by product category."*

**Database: Product**

| category | sales | region | date | ... |
|---|---|---|---|---|
| X | 5M | North | 2024-01-15 | ... |
| Y | 7.5M | South | 2024-02-20 | ... |

**Bad Reasoning Step**

**Reasoning:** The query mentions "distribution" of total sales by product category . Pie and bar charts work, and a boxplot might show distribution.

**Sub-Answer:** VIS [Bar, Pie, Boxplot].

**Good Reasoning Step**

**Reasoning:** The query indicates "distribution" of total sales by product category with one value per category. Pie and bar charts are valid, but not boxplot.

**Sub-Answer:** VIS [Bar, Pie].

**Problem:** Although boxplot is good for distribution of multiple values within categories, it is unsuitable for single values per department,

## Example 4: Error at Channel Mapping Step

**Text Query**

*"Visualize sales and region data over time in a bar chart."*

**Database: Product**

| category | sales | region | date | ... |
|---|---|---|---|---|
| X | 5M | North | 2024-01-15 | ... |
| Y | 7.5M | South | 2024-02-20 | ... |

**Bad Reasoning Step**

**Reasoning:** The query mentions "sales" and "region" over time. I'll map Sales to Y, Region to X, and Date to color.

**Sub-Answer:** X region, Y sales, COLOR Date

**Good Reasoning Step**

**Reasoning:** The query specifies "sales" and "region" over time. Sales to Y, Date to X, and Region to color.

**Sub-Answer:** X date, Y sales, COLOR region

**Problem:** Although temporal column "Date" can be mapped to COLOR in a scatterplot, it should be mapped to X in a bar chart.

Figure 10: Examples of Stepwise Reasoning Errors in the nvBench 2.0 dataset, highlighting common pitfalls in the Text-to-Visualization process.

Table 6: Licenses List for Assets Used

| Asset | Usage | License |
|---|---|---|
| GPT-4o [73] | Baselines and query verification | Custom License |
| GPT-4o-mini | Baselines, metadata generation, query synthesis | Custom License |
| Qwen2.5-7B [74] | Baselines and model fine-tuning | Apache-2.0 |
| nvBench Dataset [24] | Source for data tables and seed visualizations | MIT |
| BIRD Dataset [29] | Source for additional data tables | CC BY-SA 4.0 |
| ConceptNet [64] | Semantic alias discovery in ambiguity metadata | CC BY-SA 4.0 |
| ASP Solver (Clingo) [30] | Visualization query resolution | MIT |

# F    Limitations

While nvBench 2.0 introduces significant advancements in ambiguity-aware Text2VIS benchmarking, several limitations remain that present opportunities for future research:

**Limited Coverage of Visualization-Adjacent Tasks:** Although our benchmark focuses on Text2VIS ambiguity resolution, it does not extend to related domains such as Text2SQL. The step-wise reasoning approach could potentially be adapted to handle SQL generation with ambiguous queries, particularly since visualization and database queries share similar data operations. Future work could explore the integration of both Text2VIS and Text2SQL ambiguity resolution within a unified framework.

**Restricted Chart Types and Components:** Though nvBench 2.0 includes six chart types (bar, line, pie, scatter, heatmap, and boxplot), it does not cover the full spectrum of visualization techniques. Advanced chart types like treemaps, network diagrams, geographic maps, and multi-view coordinated visualizations are not included. Additionally, the benchmark lacks support for more sophisticated chart components such as error bars, trend lines, annotations, interactive elements, and customizable legends that are often crucial for comprehensive data storytelling.

**Limited Integration with Statistical Analysis:** The current benchmark treats visualization as the primary goal rather than integrating it with deeper statistical analysis intents. Users often request visualizations to support specific analytical objectives (hypothesis testing, correlation analysis, outlier detection, clustering) that require a tighter coupling between visualization and statistical computation. Future work could expand the benchmark to include cases where visualization serves as a component within broader analytical workflows.

**Absence of Conversational Context:** nvBench 2.0 evaluates standalone queries without considering the conversational context in which they might appear. In real-world scenarios, visualization requests often occur within multi-turn dialogues where context from previous exchanges influences interpretation. The benchmark does not account for these contextual dependencies, limiting its ability to evaluate systems in realistic interactive settings where ambiguity resolution might span multiple conversational turns.

# G    Ethic Statement

This paper introduces nvBench 2.0, a novel benchmark for ambiguous Text-to-Visualization tasks, and evaluates the capabilities of LLMs in resolving visualization ambiguity. Our work is not intended to provoke anxiety, but rather to gain a better understanding of how LLMs can be leveraged to interpret ambiguous natural language queries for data visualization. This study aims to foster discussions on how visualization systems can better accommodate the inherent ambiguity in human requests, and how humans can more effectively utilize LLM-powered visualization tools.

In developing nvBench 2.0, we have ensured that the dataset does not contain sensitive or personally identifiable information. The data tables used in our benchmark are derived from public datasets (nvBench [28] and BIRD [29]) and have been carefully reviewed to exclude potentially harmful or private content.

In our experimental evaluations involving human reviewers for query verification, participants received appropriate compensation and ensured adequate rest periods between review sessions. We rigorously

Table 7: Prompt Structure for Ambiguous Text Query Synthesis

| **Prompt Structure for Ambiguous Text Query Synthesis** |
|---|
| **### Task Description:** |
| You are an intelligent assistant. You will create three text queries (command, question, and caption) based on a given data schema and action list. Each query must incorporate all information from the action list without introducing extra elements. |
| **### Process (4 steps):** |
| **# Step 1: Interpret Visualization Type.** *e.g., ...* |
| **# Step 2: Rephrase Data Columns.** *e.g., ...* |
| **# Step 3: Rephrase Data Transformations.** *e.g., ...* |
| **# Step 4: Rephrase Filter Conditions.** *e.g., ...* |
| **### Final Answer Construction:** |
| Combine the rephrased elements from steps 1-4 to create three text queries: |
| 1. Command-style: Direct instruction (*e.g.,* "Plot the total sales by month for products priced over $50") |
| 2. Question-style: Inquiry format (*e.g.,* "What was the average rating for movies released during 2020?") |
| 3. Caption-style: Declarative format (*e.g.,* "Distribution of customer count by region in a pie chart.") |
| **### Input** |
| Database: {basename} |
| Data Columns: {data_schema} |
| Data Value Examples:{data_value_example} |
| Ambiguous Column Pairs:{ambiguous_pairs} |
| Action List:{action_list} |

protect participants' personal information, ensuring that their information remains confidential and is not disclosed in this document or the GitHub repository.

Our source code and data are under GPL-3 license, and we follow the licenses of assets used in this paper, as listed in Table 6.

Table 8: Prompt Structure for Text Query Verification

| **Prompt Structure for Text Query Verification** |
|---|
| **### Task Description:** 
 You are a smart assistant. Your job is to check if a text query for a Text-to-Visualization (Text2VIS) task is valid. The query must match all parts of the provided visualization tree (T) and follow three rules: 
 1. **Completeness**: The query must include all actions and elements from the visualization tree, such as data selections, transformations, chart types, and channel mappings. 
 2. **Type Preservation**: The query must keep the same ambiguity types (e.g., Data Transformation, Channel Mapping, Data Selection, Chart Type Selection) as defined in the visualization tree. 
 3. **Boundedness**: The query must not add extra information or elements not in the visualization tree. 
 You will get the data schema, action list, visualization tree (T), and the text query (q). Verify if the query meets all three rules. If the query is invalid, explain what is missing or extra to help fix it. |
| **### Input:** 
 # **Database**: [baseline] 
 # **Data Schema**: {data_schema} 
 # **Data Value Examples**: {data_value_example} 
 # **Visualization Tree (T)**: {vis_tree} 
 # **Synthesized Text Query (q)**: [text_query] |
| **### Output:** 
 # **Validity**: [Valid/Invalid] 
 # **Completeness**: [Met/Not Met] - Confirm if the query includes all actions and elements from the visualization tree (T). 
 # **Type Preservation**: [Met/Not Met] - Confirm if the query preserves the ambiguity types as defined in the visualization tree. 
 # **Boundedness**: [Met/Not Met] - Confirm if the query avoids adding extra information not in the visualization tree. 
 # **Feedback (if Invalid)**: Explain what makes the query invalid, including missing elements or extra information, to guide fixing or regenerating the query. |

Table 9: Prompt Structure for Step-wise Reasoning Synthesis

| Prompt Structure for Step-wise Reasoning Synthesis |
|---|
| **### Task Description:** |
| You are a good data visualization expert. Given an ambiguous/incomplete Text Query with Data Schema, and the step-by-step answer recommending visualization charts corresponding to the ambiguous/incomplete Text Query. Then, you need to fill in the reasoning process for each step. |
| **### Instructions:** |
| **# Step 1: Data Selection.** Select data columns and data filters mentioned in the Text Query. If the Text Query is ambiguous and can be mapped to multiple columns, use a list to indicate all columns. 
 **# Step 2: Data Transformation.** Select data transformation (operation : parameter) = (aggregate : [sum, mean, count]; bin : base; sort:[ascending, descending]) mentioned in the Text Query. 
 **# Step 3: Chart type Selection.** Select all valid chart types for visualization based on Text Query and data selected. If chart type indicated in the Text Query, select from chart mark=(bar, line, arc, point, rect, boxplot). Else if no chart type mentioned, but specific analysis task mentioned in the Text Query, inference chart type by (task:chart)=(trend:[bar,line]; distribution:[bar,arc,line,boxplot])... Also consider if the chart type can visualize selected data. 
 **# Step 4: Selected Column-Channel Mapping.** Map selected data columns to encoding channels = (x, y, color, size). You should consider basic channel mapping feasibility. Answer with all valid chart-channel-column mapping solutions. 
 **# Step 5: Visualization Synthesis.** Based on previous steps, synthesis the final visualizations. |
| **### Input:** |
| Database: {basename} 
 Data Columns: {data_schema} 
 Data Value Examples:{data_value_example} 
 Text Query: {text_query} |
| **### Output** |
| # Step 1: <reasoning>...</reasoning> <answer> ... <answer> 
 # (Step 2-4) 
 # Step 5: <answer> [ VIS 1, VIS 2, ..., VIS k ] <answer> |

Table 10: Prompt Structure for Basic Experiments

| Prompt Structure for Basic Experiments |
|---|
| **### Task Description:** |
| You are a good data visualization expert. Given an ambiguous/incomplete Natural Language Query and a Data Table, please recommend 1 to 5 different charts corresponding for the ambiguous/incomplete NL Query. Please strictly follow the output format. |
| **### Example:** 
 **# Input:** 
 Database: {basename} 
 Data Columns: {data_schema} 
 Data Value Examples: {data_value_example} 
 Query: {text_query} 
 **# Output:** 
 <answer> [ VIS 1, VIS 2, ..., VIS k ] </answer> |

Table 11: Prompt Structure for Stepwise Reasoning Experiments

| **Prompt Structure for Stepwise Reasoning Experiments** |
|---|
| **### Task Description:** 
 You are a good data visualization expert. Given an ambiguous/incomplete Natural Language Query and a Data Table, please recommend 1 to 5 different charts corresponding for the ambiguous/incomplete NL Query. 
 Please think step by step and strictly follow the output format. |
| **### Example:** 
 **# Input:** 
 Database: {basename} 
 Data Columns: {data_schema} 
 Data Value Examples: {data_value_example} 
 Query: {text_query} 
 **### Output:** 
 # Step 1: <reasoning>...</reasoning> <answer> ... </answer> 
 # (Step 2-4) 
 # Step 5: <answer> [ VIS 1, VIS 2, ..., VIS k ] </answer> |

