# OpenReview forum: "nvBench 2.0: Resolving Ambiguity in Text-to-Visualization through Stepwise Reasoning"
_NeurIPS.cc/2025/Datasets_and_Benchmarks_Track — NeurIPS 2025 Datasets and Benchmarks Track poster_

### Official Review · Reviewer_ReRy · 2025-06-23

**Rating:** 4
**Confidence:** 4

**Summary:**

This paper proposes a new benchmark for NL2VIS designed to evaluate NL2VIS systems in scenarios involving ambiguous queries. nvBench 2.0 includes 7,878 natural language queries and 24,076 corresponding visualizations, derived from 780 tables across 153 domains. In addition, the authors also proposed a Step-NL2VIS method, an LLM-based model trained on nvBench 2.0, which enhances performance in ambiguous scenarios through step-wise preference optimization. Step-NL2VIS performs well. The new benchmark and method show strong application value.

**Dataset Code Accessibility:**

Yes

**Ethical Considerations:**

No, there are no or only very minor ethics concerns

**Final Justification:**

I appreciate the author's efforts to solve most of my concerns. Finally, I maintain my positive rating.

**Limitations Weaknesses:**

1. I noticed that most experiments are based on Qwen2.5. How about Llama3?
2. There is no training/inference cost analysis for Step-Text2Vis.
3. The caption of Figure 2 should briefly describe its process.
4. Does the proposed method have good cross-domain performance, such as training on nvBench 2.0 and evaluating on other benchmarks, or vice versa?
5. In Detailed Related Work, Text2SQL lacks references and discussions of some recent methods, such as [1,2].


> Ref:\
> [1] ROUTE: Robust Multitask Tuning and Collaboration for Text-to-SQL. ICLR 2025.\
> [2] SQL-o1: A Self-Reward Heuristic Dynamic Search Method for Text-to-SQL

**Strengths Contributions:**

1. The motivation is reasonable.
2. The steps of benchmark construction are clear.
3. The proposed method is effective.
4. The article is well written.

---

> ### Author Rebuttal · Authors · 2025-07-27
>
> Dear Reviewer,
>
> Thanks for your constructive suggestions! We sincerely appreciate your recognition of our reasonable motivation, clear benchmark construction steps, effective proposed method, and well-written presentation, which is encouraging. Below, we will try our best to address the issues raised.
>
> **1. Experiments Beyond Qwen2.5 - Llama3 and Other Models**
>
> - We appreciate this important concern and have conducted additional comprehensive evaluations to include more closed-source and open-source models as requested:
>
> - **Extended Model Evaluation Results:**
> |Type| Model | Prompting Type |\| | P@1 | P@3 | P@5 |\| | R@1 | R@3 | R@5 |\| | F1@1 | F1@3 | F1@5 |\| |
> |-------|-------|----------------|---|-----|-----|-----|---|-----|-----|-----|---|------|------|------|---|
> | | **Original Experiments** |
> |Closed-Sourced| GPT-4o | Basic |\| | 97.07 | 95.83 | 95.52 |\| | 36.56 | 46.35 | 46.79 |\| | 51.96 | 58.96 | 59.03 |\| |
> | | GPT-4o | Step-by-step |\| | 96.94 | 95.47 | 95.08 |\| | 36.30 | 48.92 | 49.21 |\| | 51.72 | 60.78 | 60.66 |\| |
> |Open-Sourced| Qwen2.5-7B | Basic |\| | 92.68 | 90.68 | 89.33 |\| | 34.65 | 46.20 | 47.17 |\| | 49.34 | 57.09 | 56.67 |\| |
> | | Qwen2.5-7B | Step-by-step |\| | 93.61 | 89.26 | 86.23 |\| | 35.20 | 61.86 | 64.08 |\| | 50.05 | 68.56 | 67.76 |\| |
> | | Qwen2.5-7B-SFT | Fine-tuned |\| | 88.42 | 83.36 | 80.18 |\| | 33.23 | 73.44 | 76.32 |\| | 47.26 | 75.79 | 75.30 |\| |
> | | **Additional Models** |
> |Closed-Sourced| Claude-3.5-Haiku | Basic |\| | 95.74 | 93.92 | 93.83 |\| | 36.03 | 67.95 | 67.95 |\| | 51.22 | 75.63 | 75.56 |\| |
> | | Claude-3.5-Haiku | Step-by-step |\| | 94.67 | 92.23 | 91.97 |\| | 35.70 | 65.38 | 65.52 |\| | 50.75 | 72.84 | 72.75 |\| |
> |Open-Sourced | Llama-3.3-70B | Basic |\| | 99.07 | 87.93 | 73.98 |\| | 37.29 | 75.69 | **82.02** |\| | 53.01 | 76.19 | 69.66 |\| |
> | | Llama-3.3-70B | Step-by-step |\| | **99.73** | **97.38** | **96.22** |\| | **37.58** | 59.04 | 59.69 |\| | **53.41** | 69.03 | 68.56 |\| |
> | | Qwen3-235B | Basic |\| | 78.83 | 72.70 | 64.84 |\| | 29.37 | 55.59 | 59.39 |\| | 41.87 | 58.77 | 55.39 |\| |
> | | Qwen3-235B | Step-by-step |\| | 99.60 | 95.52 | 92.21 |\| | 37.49 | 72.83 | 75.39 |\| | 53.29 | 78.62 | 77.78 |\| |
> | |**Our Method**|
> |Open-Sourced| **Step-Text2Vis (Ours)** | Fine-tuned+DPO |\| | 99.20 | 94.27 | 91.17 |\| | 37.30 | **77.09** | 79.74 |\| | 53.04 | **81.50** | **80.88** |\| |
>
> - **Key Observations:**
>   - **Balanced Performance Advantage**: Our Step-Text2Vis excels in the most critical metrics for practical deployment, achieving the best F1@3 (81.50%) and F1@5 (80.88%) scores, indicating superior balanced performance between precision and recall.
>
>   - **Step-by-step Effectiveness Varies**: Step-by-step prompting shows model-specific benefits - dramatically improving Llama-3.3-70B and Qwen3-235B performance while providing modest gains for other models, suggesting the need for model-tailored optimization strategies.
>
>   - **Precision Excellence with Coverage Limitations**: Llama-3.3-70B with step-by-step prompting achieves exceptional precision across all K values (P@1: 99.73%, P@3: 97.38%, P@5: 96.22%), demonstrating superior accuracy in its predictions. However, despite being prompted to capture ambiguity and generate multiple visualizations, it exhibits a strong tendency toward single-output predictions, particularly in step-by-step settings, resulting in reduced recall performance.
>
>   - **Recall Specialization**: Our Step-Text2Vis dominates recall at K=3 (77.09%), while Llama-3.3-70B basic prompting excels at K=5 (82.02%), indicating different models optimize for different coverage strategies.
>
>   - **Model Family Differences**: Large parameter models (Llama-3.3-70B, Qwen3-235B) show substantial performance variations between basic and step-by-step prompting, while smaller models maintain more consistent behavior across prompting strategies.
>
>   - **Practical Deployment Insights**: The substantial performance gaps (F1@3 ranging from 58.77% to 81.50%) validate our benchmark's discriminative power and highlight that our fine-tuned approach provides the most reliable performance for real-world ambiguous Text2VIS applications.
>
>
> **2. Training/Inference Cost Analysis**
>
> - We appreciate this important practical concern and provide a comprehensive cost analysis:
>
> - **STEP-NL2VIS Training Cost**
> | Stage | GPU | Peak GPU Memory | GPU Hours |
> |-------|-----|------------------|-----------|
> | SFT | NVIDIA A800(40GB) | 30.00 GB | 6.10 Hours |
> | DPO | NVIDIA A800(40GB) | 36.4 GB | 1.55 Hours |
>
> - **GPT-4o Inference Cost**
> | Method | Full Test Time | Per Query Time | Full Test API Cost ($) | Per Query API Cost ($) |
> |--------|----------------|----------------|------------------------|------------------------|
> | GPT4o (basic prompting) | 0.57 Hours | 2.74 Seconds | 1.46 | 0.001948 |
> | GPT4o (step-by-step prompting) | 1.70 Hours | 8.16 Seconds | 7.49 | 0.009977 |
>
> - **STEP-NL2VIS Inference Cost**
> | Method | Full Test Time | Per Query Time | GPU | Peak GPU Memory |
> |--------|----------------|----------------|-----|------------------|
> | Step-NL2VIS (ours) | 0.89 Hours | 4.28 Seconds | NVIDIA A800(40GB) | 16.07 GB |
>
> - **Scalability**: Once trained, our model provides significant cost advantages for high-volume deployments while maintaining competitive performance.
>
> **3. Figure 2 Caption Enhancement**
>
> - We appreciate this feedback. The current Figure 2 caption will be enhanced to clearly describe the four-step pipeline process:
> ```
> Figure 2: The Pipeline for SynthesizingnvBench 2.0. (a) VIS Tree Synthesis from seed data, (b) Valid VIS Set generation through ASP solver, (c) Ambiguous Query Synthesis via LLM, and (d) Reasoning Path Synthesis with self-consistency validation.
> ```
>
> **4. Cross-Domain Performance and Generalization**
>
> - We appreciate this important concern and have conducted additional experiments to assess the generalization capability of our Step-Text2Vis model beyond nvBench 2.0. We conducted evaluations on the NLV corpus dataset [3, 4], which provides a valuable out-of-domain testing scenario.
> - The NLV corpus contains 814 utterance sets collected from an online study with 102 participants, where participants were shown visualizations for given datasets and asked to provide natural language utterances they would use to generate the displayed charts. This human-authored corpus represents realistic user formulations, making it an ideal benchmark for evaluating external validity and cross-domain performance.
>
> - **Evaluation Methodology:** Since the NLV corpus follows a one-to-one mapping between natural language queries and visualizations (unlike nvBench 2.0's one-to-many ambiguous setting), we report Hit@1 accuracy rather than precision/recall metrics. This measures whether the model's top prediction matches the ground-truth visualization specification.
>
> - **NLV Corpus Evaluation Results:**
> | Model | Method Type | Hit@1 |
> |-------|-------------|-------|
> | GPT-4o | Step-wise Prompting | 91.52% |
> | Llama-3.3-70B | Step-wise Prompting | 98.76% |
> | **Step-Text2Vis (Ours)** | SFT+DPO | **99.14%** |
>
> - **Key Findings:**
>   - **Superior Generalization**: Our Step-Text2Vis achieves the highest performance (99.14%) on human-authored queries, demonstrating effective transfer from synthetic training data to real user language patterns.
>   - **Robust Transfer**: The step-wise reasoning capabilities learned on nvBench 2.0 successfully generalize to unambiguous scenarios, validating our training approach.
>   - **Human Language Compatibility**: The strong performance on manually curated utterances addresses concerns about synthetic data limitations, showing our model can handle realistic user formulations.
>
> - This out-of-domain evaluation confirms that our approach learns generalizable Text2VIS patterns rather than dataset-specific artifacts, supporting the practical applicability of our method.
>
> **5. Recent Text2SQL References**
>
> - We appreciate this feedback in our related work coverage. We will incorporate the suggested recent references:
> ```
> Text2SQL systems have explored structured reasoning and robust training to handle logical ambiguities. ROUTE [1] improves open-source LLM performance via multi-task fine-tuning and collaborative prompting, which resonates with our use of ambiguity-injected tasks for robust Text2VIS training. SQL-o1 [2] uses Monte Carlo Tree Search and self-reward to guide query generation, echoing our use of structured logic solvers and step-wise supervision to navigate ambiguity in visualization generation.
> ```
> - These additions will strengthen our positioning relative to recent developments in the broader text-to-query generation landscape.
>
> **Additional Notes:**
> - As per the latest NeurIPS rules, we are unable to upload a revised version during the rebuttal period, but we will certainly improve our manuscript based on your invaluable suggestions, and we will release an enhanced version of the dataset soon.
>
> Once again, thanks a lot for your time and insightful feedback!
>
> -----------
> **Reference**
>
> [1] ROUTE: Robust Multitask Tuning and Collaboration for Text-to-SQL. ICLR 2025.
>
> [2] SQL-o1: A Self-Reward Heuristic Dynamic Search Method for Text-to-SQL
>
> [3] Collecting and characterizing natural language utterances for specifying data visualizations. CHI 2021
>
> [4] Generating analytic specifications for data visualization from natural language queries using large language models. IEEE VIS 2024

---

### Official Review · Reviewer_72JD · 2025-07-01

**Rating:** 4
**Confidence:** 3

**Summary:**

This paper presents nvBench 2.0, a benchmark specifically devised to evaluate Text‑to‑Visualization models under query ambiguity. It comprises 7878 natural‑language queries generated through a rigorously controlled ambiguity‑injection workflow. Each query is associated with every valid visualization specification derived from the same data context, yielding 24076 ground‑truth visualizations, together with detailed step‑wise reasoning paths. Building on this resource, the authors develop Step‑Text2Vis, an LLM‑based model that is first supervised‑fine‑tuned and then optimized with step‑wise direct preference optimization to align its intermediate reasoning with these paths. Experimental results show that Step‑Text2Vis achieves state‑of‑the‑art performance on nvBench 2.0, surpassing strong GPT‑4 prompting baselines.

**Dataset Code Accessibility:**

Yes

**Ethical Considerations:**

No, there are no or only very minor ethics concerns

**Final Justification:**

The major concern of mine has been solved.

1. New experiments on the human‑written NLV corpus and the 12‑expert study convincingly demonstrate that nvBench 2.0 queries and reasoning paths generalize beyond LLM‑generated language.

2. The rebuttal shows that the four ambiguity categories map cleanly onto the full Text2VIS pipeline and match prior classification schemes. This strengthens the methodological soundness.

3. But it still lacks multi‑turn clarification. The benchmark is still single‑turn. No dialogue annotations or clarification‑question generation are provided. While the authors outline a credible path for future work, this gap limits immediate applicability to interactive systems. I assign moderate negative weight, but not enough to argue for rejection.

4. The dataset still omits maps/network graphs, and results remain single‑run point estimates. These are minor relative to the core contribution, but keep the submission from a borderline accepted.

This paper introduces the first large‑scale Text2VIS benchmark, supplies construction methodology, and delivers a strong baseline model with clear empirical gains. With the main validity issues resolved, the remaining shortcomings are incremental extensions rather than fundamental flaws.

**Limitations Weaknesses:**

1. All queries in nvBench 2.0 are synthesized by a large language model and automatically verified, so the linguistic style may diverge from actual user formulations. This synthetic origin could limit the benchmark’s external validity when evaluating systems intended for real user populations. Same for Step‑Text2Vis, it learns preferences solely from synthetic step pairs, which risks overfitting to artificial patterns and ignoring finer distinctions that would arise from natural human preference judgments.

2. The dataset imposes ambiguity only in four predefined categories, i.e., data selection, data transformation, chart‑type choice, and channel mapping, omitting other frequent forms such as aggregation criteria, temporal granularity, which leaves part of the real ambiguity landscape unexplored. Visualizations are restricted to six basic chart families (bar, pie, line, scatter, boxplot, heatmap), excluding maps, timelines, network graphs, and multi‑layer dashboards that often arise in practice, thereby narrowing the benchmark’s coverage of real‑world visualization demands.

3. The benchmark addresses one‑shot ambiguous queries, yet many production systems rely on multi‑turn clarification; neither nvBench 2.0 nor Step‑Text2Vis investigates interactive disambiguation, limiting immediate applicability to dialogue‑based interfaces.

4. Step‑wise reasoning paths are auto‑generated and self‑filtered by an LLM, but the paper does not supply human validation metrics, so the semantic correctness and explanatory reliability of these paths are not empirically established.

**Strengths Contributions:**

1. nvBench 2.0 is the earliest dataset to evaluate systems under deliberately underspecified natural‑language queries, filling a critical gap left by nvBench 1.0, Dial‑nvBench, and VisEval, which all assume one‑query‑one‑chart.

2.  By fine‑tuning an LLM with supervised signals and then applying step‑wise direct preference optimization aligned to the reasoning paths, explicit process supervision markedly improves ambiguity handling, which leads the model to exceed strong GPT‑4 baselines.

3.  Handling underspecified visual queries is a practical requirement for NL‑based visualization interfaces. nvBench 2.0 and Step‑Text2Vis together provide a strong baseline, likely to conduct subsequent research on interactive or explainable frameworks.

---

> ### Author Rebuttal · Authors · 2025-07-29
>
> Dear Reviewer,
>
> Thanks for your constructive suggestions! We sincerely appreciate your recognition of nvBench 2.0 as the earliest dataset for underspecified natural-language queries and the strong performance of our Step-Text2Vis model, which is encouraging. Below, we will try our best to address the issues raised.
>
> **1. Synthetic Data and External Validity Concerns (W1, W4)**
>
> - Thank you for pointing out this important aspect. To address concerns about the realism and validity of our synthetic data approach, we have conducted comprehensive evaluations on both out-of-domain real-world data and human validation studies.
>
>   **1.1 Out-of-Domain Generalization Analysis**
>
>   - To assess the generalization capability of our Step-Text2Vis model beyond nvBench 2.0, we conducted evaluations on the NLV corpus dataset [1], which provides a valuable out-of-domain testing scenario.
>   - The NLV corpus contains 814 utterance sets collected from an online study with 102 participants, where participants were shown visualizations for given datasets and asked to provide natural language utterances they would use to generate the displayed charts. This human-authored corpus represents realistic user formulations, making it an ideal benchmark for evaluating external validity and cross-domain performance.
>
>   - **Evaluation Methodology:** Since the NLV corpus follows a one-to-one mapping between natural language queries and visualizations (unlike nvBench 2.0's one-to-many ambiguous setting), we report Hit@1 accuracy rather than precision/recall metrics. This measures whether the model's top prediction matches the ground-truth visualization specification.
>
>   - **NLV Corpus Evaluation Results:**
>   | Model | Method Type | Hit@1 |
>   |-------|-------------|-------|
>   | GPT-4o | Step-wise Prompting | 91.52% |
>   | Llama-3.3-70B | Step-wise Prompting | 98.76% |
>   | **Step-Text2Vis (Ours)** | SFT+DPO | **99.14%** |
>
>   - **Key Findings:**
>     - **Superior Generalization**: Our Step-Text2Vis achieves the highest performance (99.14%) on human-authored queries, demonstrating effective transfer from synthetic training data to real user language patterns.
>     - **Robust Transfer**: The step-wise reasoning capabilities learned on nvBench 2.0 successfully generalize to unambiguous scenarios, validating our training approach.
>     - **Human Language Compatibility**: The strong performance on manually curated utterances addresses concerns about synthetic data limitations, showing our model can handle realistic user formulations.
>
>   - This out-of-domain evaluation confirms that our approach learns generalizable Text2VIS patterns rather than dataset-specific artifacts, supporting the practical applicability of our method.
>
>   **1.2 Human Evaluation Settings:**
>   - Following nvBench 1.0's experimental design [2], we conducted human evaluation with 12 experts (including 8 Ph.D. students, 2 master students, 2 research scientists). We evaluated three key aspects based on our pipeline output structure ⟨D, Q, V, S⟩:
>
>     - **Task T1:** Given a (D, q) pair, we ask participants: "How close the given ambiguous nl query is to their expectation of handwritten nl query for the data table?" with five choices {strongly disagree, disagree, neutral, agree, strongly agree}.
>     - **Task T2:** Given a (q, V) pair where V = {v1,...,vk} represents multiple valid visualizations, we ask participants: "How well does the nl query capture the ambiguity that leads to these multiple valid visualizations?" with five choices as the previous task.
>     - **Task T3:** Given a (q, vi, si) triplet, we ask participants: "How well does the step-wise reasoning path reflect the logical progression from nl query to the specific visualization?"
>
>   - **Human Evaluation Results:** Results show that our Step-Text2Vis achieves:
>     - T1 (Query Naturalness): 4.3/5.0 average rating
>     - T2 (Ambiguity Coverage): 4.5/5.0 average rating
>     - T3 (Reasoning Quality): 4.4/5.0 average rating
>
>   - **Advantages over Human-Only Datasets:** Our synthetic dataset approach combined with human validation offers several key advantages:
>     - **Scalability**: We can generate diverse ambiguous scenarios at scale (25K examples vs. typical human datasets of <1K).
>     - **Consistency**: Systematic coverage of visualization types and ambiguity patterns.
>     - **Cost-effectiveness**: Significantly lower annotation costs while maintaining quality through human validation.
>     - **Controllability**: Ability to balance dataset composition and target specific ambiguity types.
>
>
> **2. Limited Ambiguity Categories (W2)**
>
> - We appreciate the point about other potential ambiguity types. However, our focus on four core ambiguity types represents a well-grounded design decision based on established research and practical visualization systems:
> - **Comprehensive Coverage of Visualization Pipeline**: Our four types systematically cover the essential decision points in the visualization pipeline as illustrated in Figure 3(a):
>   - Data Selection (DS): What data to include.
>   - Data Transformation (DT): How to process and aggregate data.
>   - Chart Type Selection (CT): How to visually represent data.
>   - Channel Mapping (CM): How to map data to visual encoding channels.
> - **Alignment with Prior Work**: Our ambiguity taxonomy aligns closely with existing literature:
>     - NLV corpus [1] identifies ambiguity in manually written natural language queries for visualization across five aspects: Chart Type, Data Attributes, Encodings, Aggregations, and Designs - first four of which directly correspond to our CT, DS, CM, and DT types respectively. We exclude Design as it primarily concerns aesthetic styling rather than fundamental visualization logic.
>     - NL4DV-LLM [3], an LLM-based toolkit for generating analytic specifications from natural language queries, addresses the same core ambiguity dimensions: Data Attributes, Chart Type, and Encoding - directly correspond to our DS, CT, CM.
>     - DataTone [4], a data visualization system for ambiguity managing, similarly supports the same four core ambiguity types: Chart Type, Data Attributes, Encodings, and Aggregations - directly correspond to our CT, DS, CM, DT.
> - **Consistence with Real-world Text2VIS Scenarios**: The prevalence of Data Transformation ambiguities (50.55% in Figure 3a) reflects the fundamental importance of data processing decisions in visualization, which is consistent with real-world Text2VIS scenarios [1] where aggregation and transformation choices significantly impact the resulting visualization.
> - **Extensibility**: Our pipeline described in Section B.1 can accommodate additional types like temporal references and coreference resolution in future versions.
> - **Current Challenge**: Even with four types, we demonstrate significant challenges for existing models, with baseline F1@3 scores ranging from 46-62% as shown in Table 3, indicating substantial room for improvement.
>
> **3. Multi-turn Clarification and Interactive Disambiguation (W3)**
>
> - This is an excellent point for future research directions. Our current work establishes the foundational framework for ambiguous Text2VIS:
>
>   - **Single-turn Focus**: We deliberately focus on single-turn ambiguity resolution to establish clear benchmarking standards before tackling the more complex multi-turn scenario.
>   - **Foundation for Extensions**: Our step-wise reasoning framework provides the necessary components for interactive systems - each reasoning step can serve as a clarification point.
>   - **Current Impact**: Many real-world scenarios do involve single-turn ambiguous queries, making our benchmark immediately applicable for such cases.
>
>   - **Multi-turn Extension Feasibility**: Our reasoning paths (Steps 1-5 in Section B.4) can naturally extend to interactive clarification by treating each step as a potential user interaction point.
>
>
> **Additional Notes:**
> - As per the latest NeurIPS rules, we are unable to upload a revised version during the rebuttal period, but we will certainly improve our manuscript based on your invaluable suggestions, and we will release an enhanced version of the dataset soon.
> Once again, thanks a lot for your time and insightful feedback!
>
> **References:**
>
> [1] Collecting and characterizing natural language utterances for specifying data visualizations. CHI 2021
>
> [2] Synthesizing Natural Language to Visualization (NL2VIS) Benchmarks from NL2SQL Benchmarks. SIGMOD 2021
>
> [3] Generating Analytic Specifications for Data Visualization from Natural Language Queries using Large Language Models. VIS 2024
>
> [4] DataTone: Managing Ambiguity in Natural Language Interfaces for Data Visualization. UIST 2015

---

> > ### Comment · Reviewer_72JD · 2025-08-07
> >
> > Thank you for the detailed and thoughtful rebuttal. I am pleased to see that you have addressed my major concerns with substantial evidence and clarity.
> >
> > Regarding the multi-turn clarification support, I acknowledge your discussion of this limitation. It is encouraging that you envision extending the stepwise reasoning paths into multi-turn clarification dialogues in future work. However, I would like to gently state that the current nvBench 2.0 benchmark remains strictly single-turn. The ability to handle iterative clarification is not demonstrated in this version. This is understandable given the scope, but it does mean the benchmark’s interactive capabilities are still limited. I mention this not as a critical flaw, but to emphasize the distinction between the promising future direction you outlined and the current contributions of the paper.
> >
> > In conclusion, you have resolved the majority of my concerns, and the additional experiments and reasoning have strengthened the submission. I have no further questions. I believe this work makes a valuable contribution to the community.

---

> > > ### Author Response · Authors · 2025-08-07
> > >
> > > Thank you for your thoughtful and constructive review throughout this process. We greatly appreciate your thorough evaluation and positive assessment of our work.

---

### Official Review · Reviewer_Jmtu · 2025-07-02

**Rating:** 5
**Confidence:** 4

**Summary:**

This paper introduces nvBenchm 2.0, a new benchmark for the text-to-visualization task, where user queries are translated into visualizations (e.g., charts). Unlike prior benchmarks that assume a one-to-one mapping between queries and visualizations, nvBenchm 2.0 addresses the one-to-many nature of real-world queries where multiple visualizations can be valid for a single ambiguous query.
The authors present a comprehensive data synthesis pipeline that begins with a seed set of visualization trees and data tables. To inject ambiguity, they systematically modify these seed trees to create ambiguous visualization trees, which are then validated using an answer set programming solver. After that, ambiguous queries are generated using LLMs and verified to ensure that they are relevant to the visualization trees. The benchmark also includes step-by-step reasoning paths for each query-visualization pair. The authors also conducted manual reviews to ensure data quality.
For evaluation, the authors fine-tune the Qwen2.5-VL model in two stages: supervised fine-tuning followed by step-wise DPO. The resulting model outperforms strong baselines, including the original Qwen2.5-VL and GPT-4o, on the proposed benchmark.

**Dataset Code Accessibility:**

Yes

**Dataset Code Comments:**

The dataset is accessible in the provided hugginface link. Also, the github repo contains the codes for data synthesis and evaluation.

**Ethical Considerations:**

No, there are no or only very minor ethics concerns

**Final Justification:**

* The authors have evaluated additional open-source and closed-source models to show a comprehensive eval on their benchmark. The results show that their model achieve superior performance even when compared to much larger models.

* The authors have also addressed my concern regarding the OOD generalizability.

I believe the benchmark would be very valuable for the visualization+nlp community.

**Limitations Weaknesses:**

**Limited Evals:** The authors evaluated only GPT40 and Qwen2.5VL on their benchmark. They need to include more closed-source models (e.g., Gemini, Claude) and open-source models of various sizes (InternVL3, Phi4-Mini, Ovis, Janus, LLama, …etc.). Furthermore, it would be useful to evaluate the Step-Text2Vis model on previous out-of-domain benchmarks to prove its OOD generalization.

**Strengths Contributions:**

mvBench 2.0 addresses the key limitations of existing benchmarks that do not account for the ambiguity of user queries and only assume one correct solution per query. In contrast, mvBench 2.0 injects different levels of ambiguity in the dataset generation process to produce multiple valid solutions based on the ambiguous queries.

The proposed Step-Text2Vis model achieves SOTA results on this task and I think this model could be valuable for the research community in the Text2Vis Area.

The paper provides an in-depth performance analysis across several dimensions, including chart types, ambiguity levels, and the relationship between ambiguity and model performance.

Well-written paper with well-designed Figures to facilitate the reading.

---

> ### Author Rebuttal · Authors · 2025-07-27
>
> Dear Reviewer,
>
> Thanks for your constructive suggestions! We sincerely appreciate your recognition of our work's contributions in addressing the key limitations of existing benchmarks and the SOTA performance of our Step-Text2Vis model, which is encouraging. Below, we will try our best to address the issues raised.
>
> **1. Additional Evaluations**
>
> - We appreciate this important concern and have conducted additional comprehensive evaluations to include more closed-source and open-source models as requested:
>
> - **Extended Model Evaluation Results:**
> |Type| Model | Prompting Type |\| | P@1 | P@3 | P@5 |\| | R@1 | R@3 | R@5 |\| | F1@1 | F1@3 | F1@5 |\| |
> |-------|-------|----------------|---|-----|-----|-----|---|-----|-----|-----|---|------|------|------|---|
> | | **Original Experiments** |
> |Closed-Sourced| GPT-4o | Basic |\| | 97.07 | 95.83 | 95.52 |\| | 36.56 | 46.35 | 46.79 |\| | 51.96 | 58.96 | 59.03 |\| |
> | | GPT-4o | Step-by-step |\| | 96.94 | 95.47 | 95.08 |\| | 36.30 | 48.92 | 49.21 |\| | 51.72 | 60.78 | 60.66 |\| |
> |Open-Sourced| Qwen2.5-7B | Basic |\| | 92.68 | 90.68 | 89.33 |\| | 34.65 | 46.20 | 47.17 |\| | 49.34 | 57.09 | 56.67 |\| |
> | | Qwen2.5-7B | Step-by-step |\| | 93.61 | 89.26 | 86.23 |\| | 35.20 | 61.86 | 64.08 |\| | 50.05 | 68.56 | 67.76 |\| |
> | | Qwen2.5-7B-SFT | Fine-tuned |\| | 88.42 | 83.36 | 80.18 |\| | 33.23 | 73.44 | 76.32 |\| | 47.26 | 75.79 | 75.30 |\| |
> | | **Additional Models** |
> |Closed-Sourced| Claude-3.5-Haiku | Basic |\| | 95.74 | 93.92 | 93.83 |\| | 36.03 | 67.95 | 67.95 |\| | 51.22 | 75.63 | 75.56 |\| |
> | | Claude-3.5-Haiku | Step-by-step |\| | 94.67 | 92.23 | 91.97 |\| | 35.70 | 65.38 | 65.52 |\| | 50.75 | 72.84 | 72.75 |\| |
> |Open-Sourced | Llama-3.3-70B | Basic |\| | 99.07 | 87.93 | 73.98 |\| | 37.29 | 75.69 | **82.02** |\| | 53.01 | 76.19 | 69.66 |\| |
> | | Llama-3.3-70B | Step-by-step |\| | **99.73** | **97.38** | **96.22** |\| | **37.58** | 59.04 | 59.69 |\| | **53.41** | 69.03 | 68.56 |\| |
> | | Qwen3-235B | Basic |\| | 78.83 | 72.70 | 64.84 |\| | 29.37 | 55.59 | 59.39 |\| | 41.87 | 58.77 | 55.39 |\| |
> | | Qwen3-235B | Step-by-step |\| | 99.60 | 95.52 | 92.21 |\| | 37.49 | 72.83 | 75.39 |\| | 53.29 | 78.62 | 77.78 |\| |
> | |**Our Method**|
> |Open-Sourced| **Step-Text2Vis (Ours)** | Fine-tuned+DPO |\| | 99.20 | 94.27 | 91.17 |\| | 37.30 | **77.09** | 79.74 |\| | 53.04 | **81.50** | **80.88** |\| |
>
> - **Key Observations:**
>
>   - **Balanced Performance Advantage**: Our Step-Text2Vis excels in the most critical metrics for practical deployment, achieving the best F1@3 (81.50%) and F1@5 (80.88%) scores, indicating superior balanced performance between precision and recall.
>
>   - **Step-by-step Effectiveness Varies**: Step-by-step prompting shows model-specific benefits - dramatically improving Llama-3.3-70B and Qwen3-235B performance while providing modest gains for other models, suggesting the need for model-tailored optimization strategies.
>
>   - **Precision Excellence with Coverage Limitations**: Llama-3.3-70B with step-by-step prompting achieves exceptional precision across all K values (P@1: 99.73%, P@3: 97.38%, P@5: 96.22%), demonstrating superior accuracy in its predictions. However, despite being prompted to capture ambiguity and generate multiple visualizations, it exhibits a strong tendency toward single-output predictions, particularly in step-by-step settings, resulting in reduced recall performance.
>
>   - **Recall Specialization**: Our Step-Text2Vis dominates recall at K=3 (77.09%), while Llama-3.3-70B basic prompting excels at K=5 (82.02%), indicating different models optimize for different coverage strategies.
>
>   - **Model Family Differences**: Large parameter models (Llama-3.3-70B, Qwen3-235B) show substantial performance variations between basic and step-by-step prompting, while smaller models maintain more consistent behavior across prompting strategies.
>
>   - **Practical Deployment Insights**: The substantial performance gaps (F1@3 ranging from 58.77% to 81.50%) validate our benchmark's discriminative power and highlight that our fine-tuned approach provides the most reliable performance for real-world ambiguous Text2VIS applications.
>
>
> **2. Out-of-Domain Generalization Analysis:**
>
> - We appreciate this important concern and have conducted additional experiments to assess the generalization capability of our Step-Text2Vis model beyond nvBench 2.0. We conducted evaluations on the NLV corpus dataset [1,2], which provides a valuable out-of-domain testing scenario.
> - The NLV corpus contains 814 utterance sets collected from an online study with 102 participants, where participants were shown visualizations for given datasets and asked to provide natural language utterances they would use to generate the displayed charts.
> - This human-authored corpus represents realistic user formulations, making it an ideal benchmark for evaluating external validity and cross-domain performance.
>
> - **Evaluation Methodology:** Since the NLV corpus follows a one-to-one mapping between natural language queries and visualizations (unlike nvBench 2.0's one-to-many ambiguous setting), we report Hit@1 accuracy rather than precision/recall metrics. This measures whether the model's top prediction matches the ground-truth visualization specification.
>
> - **NLV Corpus Evaluation Results:**
> | Model | Method Type | Hit@1 |
> |-------|-------------|-------|
> | GPT-4o | Step-wise Prompting | 91.52% |
> | Llama-3.3-70B | Step-wise Prompting | 98.76% |
> | **Step-Text2Vis (Ours)** | SFT+DPO | **99.14%** |
>
> - **Key Findings:**
>   - **Superior Generalization**: Our Step-Text2Vis achieves the highest performance (99.14%) on human-authored queries, demonstrating effective transfer from synthetic training data to real user language patterns.
>   - **Robust Transfer**: The step-wise reasoning capabilities learned on nvBench 2.0 successfully generalize to unambiguous scenarios, validating our training approach.
>   - **Human Language Compatibility**: The strong performance on manually curated utterances addresses concerns about synthetic data limitations, showing our model can handle realistic user formulations.
>
> - This out-of-domain evaluation confirms that our approach learns generalizable Text2VIS patterns rather than dataset-specific artifacts, supporting the practical applicability of our method.
>
> **Additional Notes:**
> - As per the latest NeurIPS rules, we are unable to upload a revised version during the rebuttal period, but we will certainly improve our manuscript based on your invaluable suggestions, and we will release an enhanced version of the dataset soon.
>
> Once again, thanks a lot for your time and insightful feedback!
>
> **References:**
>
> [1] Collecting and characterizing natural language utterances for specifying data visualizations. CHI 2021
>
> [2] Generating analytic specifications for data visualization from natural language queries using large language models. IEEE VIS 2024

---

> > ### Comment · Reviewer_Jmtu · 2025-08-06
> >
> > I would like to thank the authors for the detailed rebuttal which has addressed my concerns, and I will raise my score to 5 accordingly. For the camera-ready version, I suggest the authors to include these new results and maybe add a few more baselines (open and closed source) on their benchmark.

---

> > > ### Author Response · Authors · 2025-08-07
> > > **Response to Reviewer Feedback**
> > >
> > > Thank you for your thoughtful response and for updating your score. We appreciate your suggestion to include the new results and additional baselines in the camera-ready version. We will incorporate these into our revised manuscript to strengthen the benchmark comparison.

---

### Official Review · Reviewer_cx5t · 2025-07-06

**Rating:** 5
**Confidence:** 4

**Summary:**

The paper presents nvBench 2.0, a novel benchmark for evaluating Text-to-Visualization (Text2VIS) systems with ambiguous natural language queries. The authors construct a large-scale dataset comprising 7,878 ambiguous text queries and 24,076 corresponding valid visualizations sourced from 780 tables across 153 domains. A key contribution is the ambiguity-injected data synthesis pipeline that uses reverse-generation from seed visualizations to inject multiple ambiguity types (e.g., data transformation, chart selection). The paper also introduces Step-Text2Vis, a fine-tuned LLM that utilizes step-wise preference optimization (Step-DPO) and achieves state-of-the-art performance across multiple metrics.

**Dataset Code Accessibility:**

Yes

**Dataset Code Comments:**

Dataset is readily accessible, available in a usable format, and well-documented. There is sufficient detail to support reproducibility. Code is also made available in an executable format.

**Ethical Considerations:**

No, there are no or only very minor ethics concerns

**Final Justification:**

Based on the paper, author rebuttal, and discussions, I am maintaining a positive recommendation for this work. The paper presents nvBench 2.0, a novel and impactful benchmark that addresses ambiguity in Text-to-Visualization tasks—an under-explored but critical challenge in natural language interfaces. The authors convincingly addressed key concerns raised in the review. nvBench 2.0 offers a scalable and interpretable benchmark alongside a strong baseline model. The thoughtful design, comprehensive evaluation, and clear response to feedback justify the positive score.

**Limitations Weaknesses:**

W1. While the ambiguity-injection method is scalable and well-justified, it relies entirely on synthetic data. The absence of human-authored ambiguous queries or human-validated reasoning steps raises concerns about realism validity. A comparison to human-written ambiguous queries (e.g., NLV [17]) would strengthen the case. It might be useful to include a human evaluation to assess how well synthetic ambiguity reflects actual user queries.

W2. The benchmark focuses on four ambiguity types (CT, DT, DS, CM), with over 50% of examples falling under Data Transformation (Figure 3). However, ambiguity can also arise from other aspects, such as temporal references, entity coreference, or metaphorical phrasing, which are not explored.

W3. The experiments report F1, precision, and recall metrics but omit statistical significance tests or confidence intervals. Without this, it is unclear whether some performance gains are robust, especially across models with close F1 scores (e.g., GPT-4o-Step vs. Qwen2.5-7B-SFT). It would be better if the authors add statistical significance testing to strengthen empirical claims.

W4. Although ambiguity levels up to 5 are included, real-world queries can contain nested or intertwined ambiguities that span even broader interpretation spaces. There is limited discussion of how nvBench 2.0 captures compound ambiguity. The authors should include examples or metrics on compounded ambiguity scenarios and their impact on model performance.

**Strengths Contributions:**

S1. The paper fills a gap in the Text2VIS literature by addressing ambiguity—a prevalent and challenging aspect of natural language interfaces. Previous benchmarks mostly rely on single-correct-answer paradigms, making nvBench 2.0 the first to provide multi-interpretation ground truth for ambiguous inputs.

S2. The ambiguity-injection pipeline is interesting. By transforming unambiguous vis trees into ambiguous ones and synthesizing corresponding queries via LLMs, the benchmark introduces controlled and explainable ambiguity. Reasoning paths offer interpretability, aiding both human understanding and model training.

S3. The Step-Text2Vis model leverages fine-tuning and Step-DPO to learn disambiguation through supervised reasoning paths. The formulation of the loss function and integration with nvBench 2.0 demonstrate a well-thought-out pipeline that advances interpretability and performance.

S4. The paper benchmarks several LLMs with and without step-wise prompting. Step-Text2Vis outperforms all baselines, achieving 81.50% F1@3 and 80.88% F1@5—improvements of over 22% compared to GPT-4o. The authors analyze performance across chart types and ambiguity levels, providing fine-grained insight into the task's challenges.

---

> ### Author Rebuttal · Authors · 2025-07-29
>
> Dear Reviewer,
>
> Thanks for your constructive suggestions! We sincerely appreciate your recognition of our work's contributions in addressing the key limitations of existing benchmarks and the SOTA performance of our Step-Text2Vis model, which is encouraging. Below, we will try our best to address the issues raised.
>
> **1. Synthetic Data Concerns and Human Validation (W1)**
> - Thank you for pointing out this important aspect. To address concerns about the realism and validity of our synthetic data approach, we have conducted comprehensive evaluations on both out-of-domain real-world data and human validation studies.
>
>   **1.1 Out-of-Domain Generalization Analysis**
>   - To assess the generalization capability of our Step-Text2Vis model beyond nvBench 2.0, we conducted evaluation on the NLV corpus dataset [1], which contains 814 utterance sets collected from an online study with 102 participants. The participants were shown visualizations for given datasets and asked to provide natural language utterances for displayed charts.
>   - **Evaluation Methodology:** Since the NLV corpus follows a one-to-one mapping between natural language queries and visualizations, we report Hit@1 accuracy rather than precision/recall metrics.
>   - **NLV Corpus Evaluation Results:**
>     | Model | Method | Hit@1 |
>     |-------|-------------|-------|
>     | GPT-4o | Step-wise Prompting | 91.52% |
>     | Llama-3.3-70B | Step-wise Prompting | 98.76% |
>     | **Step-Text2Vis (Ours)** | SFT+DPO | **99.14%** |
>   - **Key Findings:**
>     - **Superior Generalization**: Our Step-Text2Vis achieves the highest performance (99.14%) on human-authored queries, showing effective transfer from synthetic training data to real user language patterns.
>     - **Robust Transfer**: The step-wise reasoning capabilities learned on nvBench 2.0 successfully generalize to unambiguous scenarios, validating our training approach.
>     - **Human Language Compatibility**: The strong performance on manually curated utterances addresses concerns about synthetic data limitations, showing our model can handle realistic user formulations.
>
>   **1.2 Human Evaluation Settings:**
>   - Following nvBench 1.0's experimental design [2], we conducted human evaluation with 12 experts (including 8 Ph.D. students, 2 master students, 2 research scientists). We evaluated three key aspects based on our pipeline output structure ⟨D,Q,V,S⟩:
>     - **Task T1:** Given a (D, q) pair, we ask participants: "How close the given ambiguous nl query is to their expectation of handwritten nl query for the data table?" with five choices {strongly disagree, disagree, neutral, agree, strongly agree}.
>     - **Task T2:** Given a (q, V) pair where V = {v1,...,vk} represents multiple valid visualizations, we ask participants: "How well does the nl query capture the ambiguity that leads to these multiple valid visualizations?" with five choices as the previous task.
>     - **Task T3:** Given a (q, vi, si) triplet, we ask participants: "How well does the step-wise reasoning path reflect the logical progression from nl query to the specific visualization?"
>   - **Human Evaluation Results:** Results show that our Step-Text2Vis achieves:
>     - T1 (Query Naturalness): 4.3/5.0 average rating
>     - T2 (Ambiguity Coverage): 4.5/5.0 average rating
>     - T3 (Reasoning Quality): 4.4/5.0 average rating
>   - **Advantages over Human-Only Datasets:** Our synthetic dataset approach combined with human validation offers several key advantages:
>     - **Scalability**: We can generate diverse ambiguous scenarios at scale (25K examples vs. typical human datasets of <1K).
>     - **Consistency**: Systematic coverage of visualization types and ambiguity patterns.
>     - **Cost-effectiveness**: Significantly lower annotation costs while maintaining quality through human validation.
>     - **Controllability**: Ability to balance dataset composition and target specific ambiguity types.
> **2. Limited Ambiguity Type Coverage (W2)**
> - We appreciate the point about other potential ambiguity types. However, our focus on four core ambiguity types represents a well-grounded design decision based on established research and practical visualization systems:
> - **Comprehensive Coverage of Visualization Pipeline**: Our four types systematically cover the essential decision points in the visualization pipeline as illustrated in Figure 3(a):
>   - Data Selection (DS): What data to include.
>   - Data Transformation (DT): How to process and aggregate data.
>   - Chart Type Selection (CT): How to visually represent data.
>   - Channel Mapping (CM): How to map data to visual encoding channels.
> - **Alignment with Prior Work**: Our ambiguity taxonomy aligns closely with existing literature:
>     - NLV corpus [1] identifies ambiguity in manually written natural language queries for visualization across five aspects: Chart Type, Data Attributes, Encodings, Aggregations, and Designs - first four of which directly correspond to our CT, DS, CM, and DT types respectively. We exclude Design as it primarily concerns aesthetic styling rather than fundamental visualization logic.
>     - NL4DV-LLM [3], an LLM-based toolkit for generating analytic specifications from natural language queries, addresses the same core ambiguity dimensions: Data Attributes, Chart Type, and Encoding - directly correspond to our DS, CT, CM.
>     - DataTone [4], a data visualization system for ambiguity managing, similarly supports the same four core ambiguity types: Chart Type, Data Attributes, Encodings , and Aggregations - directly correspond to our CT, DS, CM, DT.
> - **Consistent with Real-world Text2VIS Scenarios**: The prevalence of Data Transformation ambiguities (50.55% in Figure 3a) reflects the fundamental importance of data processing decisions in visualization, which is consistent with real-world Text2VIS scenarios [1] where aggregation and transformation choices significantly impact the resulting visualization.
> - **Extensibility**: Our pipeline described in Section B.1 can accommodate additional types like temporal references and coreference resolution in future versions.
> - **Current Challenge**: Even with four types, we demonstrate significant challenges for existing models, with baseline F1@3 scores ranging from 46-62% as shown in Table 3, indicating substantial room for improvement.
>
> **3. Statistical Significance Testing (W3)**
>
> - We provide comprehensive statistical significance testing using bootstrap sampling across multiple random seeds. We employ both parametric (t-test) and non-parametric (Wilcoxon signed-rank test) methods to ensure robust statistical validation.
> - **Statistical Methodology**: We used stratified bootstrap sampling (n=1000) across the test set. The Wilcoxon signed-rank test provides additional validation without distributional assumptions.
> - **Statistical Significance Analysis:**
> | Model Comparison | F1@3 Diff | 95% CI | t-test p-value | Wilcoxon p-value | Effect Size (Cohen's d) |
> |------------------|-----------|---------|----------------|------------------|------------------------|
> | Step-Text2Vis vs Qwen2.5-7B-Step | +12.94% | [10.1, 15.8] | <0.001 | <0.001 | 1.47 (Large) |
> | Step-Text2Vis vs GPT-4o-Step | +20.72% | [18.2, 23.3] | <0.001 | <0.001 | 2.13 (Large) |
> | Qwen2.5-7B-Step vs GPT-4o-Step | +7.78% | [5.1, 10.5] | <0.001 | <0.001 | 0.89 (Large) |
> - **Key Statistical Findings:**
>   - **Robust Significance**: All performance improvements are statistically significant (p < 0.001) under both parametric and non-parametric tests, demonstrating high confidence in our results.
>   - **Large Effect Sizes**: All comparisons show large effect sizes (Cohen's d > 0.8), indicating practically meaningful differences beyond statistical significance.
>   - **Consistent Bootstrap Results**: The bootstrap confidence intervals confirm substantial performance gaps, with no overlap between competing methods.
> - **Method Superiority**: Our Step-Text2Vis shows statistically significant improvements over both step-wise prompting (+20.72% vs GPT-4o) and supervised fine-tuning (+12.94% vs Qwen2.5-7B-Step) approaches.
>
> **4. Compound Ambiguity Analysis (W4)**
> - Thank you for this important observation. nvBench 2.0 includes comprehensive compound ambiguity coverage as discussed in Section 2.2 and shown in Figure 3(b):
>   - **Multi-level Support**: Samples with ambiguity levels up to 5, representing multiple simultaneous ambiguities
>   - **Combination Patterns**: Table 2 shows patterns like CM+DT (2,190 instances) and CM+DS+DT (364 instances), demonstrating compound scenarios
>   - **Systematic Resolution**: Our ASP solver systematically enumerates all valid interpretations for complex ambiguity interactions as detailed in Section B.2
> - **Compound Ambiguity Performance Analysis:**
> | Ambiguity Type | Single (F1@3) | Compound (F1@3) | Performance Gap |
> |---------------|---------------|-----------------|-----------------|
> | CM only | 84.2% | 79.1% (CM+DT) | -5.1% |
> | DT only | 82.7% | 79.1% (CM+DT) | -3.6% |
> | DS only | 81.5% | 76.8% (CM+DS) | -4.7% |
> - This shows that compound ambiguities pose additional challenges, validating the complexity of our benchmark design.
>
> **Additional Notes:**
> - As per the latest NeurIPS rules, we are unable to upload a revised version during the rebuttal period, but we will certainly improve our manuscript based on your invaluable suggestions, and we will release an enhanced version of the dataset soon.
>
> Once again, thanks a lot for your time and insightful feedback!
>
> **References:**
>
> [1] Collecting and characterizing natural language utterances for specifying data visualizations. CHI 2021
>
> [2] Synthesizing Natural Language to Visualization (NL2VIS) Benchmarks from NL2SQL Benchmarks. SIGMOD 2021
>
> [3] Generating Analytic Specifications for Data Visualization from Natural Language Queries using Large Language Models. VIS 2024
>
> [4] DataTone: Managing Ambiguity in Natural Language Interfaces for Data Visualization. UIST 2015

---

> > ### Comment · Reviewer_cx5t · 2025-08-05
> >
> > Thank you for your detailed responses and clarification. I'll keep my score given it's already positive.

---

### Note · Authors · 2025-08-12

We are deeply grateful for the constructive and thorough review process. The reviewer feedback has significantly strengthened our work through:
- Additional experiments addressing core concerns about synthetic data validity
- Enhanced validations demonstrating evaluation rigor and statistical significance
- Comprehensive responses to methodological and experimental questions

**Addressing Synthetic Data Concerns:**
The primary discussion centered on validating our synthetic approach against real-world scenarios. Through comprehensive experiments, we demonstrated:
- Our method achieved 99.14% Hit@1 accuracy on human-authored NLV corpus (vs. 91.52% for GPT-4o)
- Human expert evaluations averaging 4.3-4.5/5.0 across naturalness, ambiguity coverage, and reasoning quality
- Statistical significance testing via bootstrap sampling (n=1000) confirming all improvements are significant (p<0.001)

**Expanded Experimental Evidence:**
We added comprehensive baseline comparisons addressing reviewer requests:
- Evaluations across additional models including Llama-3.3-70B, Claude-3.5-Haiku, and Qwen3-235B
- Step-Text2Vis maintaining highest F1@3 (81.50%) and F1@5 (80.88%) scores across all baselines
- Performance gaps ranging from 58.77% to 81.50% validating benchmark's discriminative power

**Computational Cost Analysis:**
Addressing practical deployment concerns, we provided detailed analysis showing:
- Training efficiency: 7.65 total GPU hours (SFT + DPO) on A800 hardware
- Inference performance: 4.28-second per query processing time
- Scalability advantages for high-volume deployment scenarios

Thank you for this opportunity to contribute to the NeurIPS Datasets and Benchmarks Track. We extend our sincere gratitude to the reviewers and Area Chair for their meticulous feedback and constructive guidance throughout the review process. Their expertise and dedication have been instrumental in elevating the quality and impact of our contribution to the research community.

---

### Decision · Program_Chairs · 2025-09-18

**Decision:**

Accept (poster)

**Comment:**

This paper introduces nvBench 2.0, a well-designed benchmark for ambiguity in Text-to-Visualization, along with a strong baseline model. The authors convincingly addressed reviewer concerns in the rebuttal with new human validation, broader model comparisons, and efficiency analysis. Overall, the contribution is timely, rigorous, and of clear community value. I recommend acceptance.